# Sleep deprivation and sleep intensity exert distinct effects on cerebral vasomotion and brain pulsations driven by the respiratory and cardiac cycles

Sara Marie Ulv Larsen[1,2☾], Sebastian Camillo Holst[1,3☾], Anders Stevnhoved Olsen[1,4], Brice Ozenne[1,5], Dorte Bonde Zilstorff[1], Kristoffer Brendstrup-Brix[1,2], Pia Weikop[6], Simone Pleinert[1], Vesa Kiviniemi[7,8], Poul Jørgen Jennum[9,10], Maiken Nedergaard[2,6,11], Gitte Moos Knudsen[1,12]*

**1** Neurobiology Research Unit, Copenhagen University Hospital, Rigshospitalet, Copenhagen, Denmark, **2** Faculty of Health and Medical Sciences (SUND), University of Copenhagen, Copenhagen, Denmark, **3** Roche Pharma Research and Early Development, Neuroscience and Rare Diseases, Roche Innovation Center Basel, Basel, Switzerland, **4** Department of Applied Mathematics and Computer Science, Technical University of Denmark, Kgs. Lyngby, Denmark, **5** Department of Public Health, Section of Biostatistics, University of Copenhagen, Copenhagen, Denmark, **6** Center for Translational Neuromedicine, University of Copenhagen, Copenhagen, Denmark, **7** Oulu Functional Neuroimaging (OFNI), Department of Diagnostic Radiology, Oulu University Hospital, Oulu, Finland, **8** Biocenter Oulu, Oulu University, Oulu, Finland, **9** Danish Center for Sleep Medicine, Department of Clinical Neurophysiology, Copenhagen University Hospital, Rigshospitalet, Glostrup, Denmark, **10** Department of Health Technology, Technical University of Denmark, Kgs Lyngby, Denmark, **11** Center for Translational Neuromedicine, Department of Neurosurgery, University of Rochester Medical Center, Rochester, New York, United States of America, **12** Department of Clinical Medicine, University of Copenhagen, Copenhagen, Denmark

☾ These authors contributed equally to this work

* gmk@nru.dk

## Abstract

The flow of cerebrospinal fluid (CSF) through the brain is driven by cerebral vasomotion, along with respiratory and cardiac forces. Growing evidence suggests that sleep facilitates this flow, yet the role of homeostatic sleep mechanisms remains largely unknown. In a circadian-controlled sleep and sleep deprivation study in humans, we used accelerated neuroimaging to investigate how sleep pressure and slow-wave-rich sleep affect low-frequency brain pulsations (LFPs; 0.012–0.034 Hz) as well as brain pulsations originating from the respiratory and cardiac cycles. These pulsations cause movement of CSF and brain tissue which may facilitate waste clearance. We also examined the origin of LFPs through pharmacological vasodilation of the cerebral vasculature with the adrenergic antagonist carvedilol in a randomized, crossover, double-blinded, placebo-controlled design (NCT03576664). We find that sleep deprivation increases LFPs more than nonrapid eye movement (NREM) sleep does, with LFPs during sleep correlating with cognitive measures of sleep pressure. Conversely, NREM sleep (combined stages N2 and N3) enhances brain pulsations driven by the respiration and cardiac cycles, with more pronounced effects in gray and white matter than in the ventricles. The strength of these brain pulsations escalates

**Data availability statement:** The data supporting the findings are not publicly available due to privacy and European General Data Protection Regulation (GDPR) restrictions. The data can be made available on request from The Center for Integrated Molecular Brain Imaging (CIMBI) database (http://www.nru.dk/cimbidb) by contacting the database manager at cimbi@cimbi.dk. A local board will review the application for data access and access to data must in most cases be granted through a data sharing agreement to be established and approved by the Danish Data Protection Agency and the legal department at Copenhagen University Hospital.

**Funding:** The research reported in this work was funded by the H2020 Marie Skłodowska-Curie Actions grant—marie-sklodowska-curie-actions.ec.europa.eu: H2020-MSCA-IF-2017-798131 (SCH); the Lundbeck Foundation grant—lundbeckfonden.com: R264-2017-2778 (SCH) & R279-2018-1145 (GMK); the Independent Research Fund Denmark grant—dff.dk: 0134-00454B & 7025-00090B (GMK); Rigshospitalet Forskningspuljer grant—rigshospitalet.dk: R151-A6534 (GMK); the EU Joint Programme—Neurodegenerative Disease Research grant—neurodegenerationresearch.eu/: 2022-120 (VK, GMK, MN). The funders had no role in research conceptualization, study design, data collection and analyses, decision to publish, or manuscript preparation.

**Competing interests:** I have read the journal's policy and the authors of this manuscript have the following competing interests: S.C.H. is currently an employee of Roche Pharma, which is unrelated to the contents of this manuscript. All other authors have declared that no competing interests exist.

**Abbreviations :** CSF, cerebrospinal fluid; ECG, electrocardiogram; EEG, electroencephalogram; EOG, electrooculogram; GM, gray matter; ICP, intracranial pressure; LFPs, low-frequency brain pulsations; MREG, Magnetic Resonance Encephalography; NREM, nonrapid eye movement; PSG, polysomnography; PVT, psychomotor vigilance test.WM, white matter.

with sleep depth (N3 > N2) and correlates with EEG delta power, a measure of slow wave activity. Moreover, carvedilol dampens LFPs, supporting that these reflect cerebral vasomotion. In summary, our findings indicate that heightened sleep pressure promotes vasomotion, whereas slow-wave-rich sleep amplifies respiration- and cardiac-driven brain pulsations, possibly indicating increased CSF flow to the brain. Together, this suggests that homeostatic sleep mechanisms are integral to human brain fluid dynamics and potentially also waste clearance.

## Introduction

Deep sleep is essential for maintaining a healthy brain function. A prominent characteristic of deep nonrapid eye movement (NREM) sleep is the presence of slow, homeostatically regulated, high-amplitude waves in the electroencephalogram (EEG) delta range (0.5–4.5 Hz) [1]. These EEG slow waves are increased by sleep deprivation [2,3] and have been implicated in multiple basic physiological processes, including brain plasticity [4], memory consolidation [5], and metabolic and immune system function [6]. Emerging preclinical evidence suggests that slow waves also play an important role in brain fluid dynamics [7], i.e., the temporal and spatial changes in cerebrospinal fluid (CSF) movement within the ventricles, subarachnoid space, and perivascular and interstitial spaces. Specifically, NREM sleep enhances the flow of CSF from perivascular spaces into and through the brain parenchyma, causing an increase in interstitial space volume and an exchange of CSF and interstitial fluid [8–10]. This influx of CSF strongly correlates with EEG slow waves in anaesthetized mice [9] and facilitates the clearance of metabolic waste from the rodent brain [8,11,12] through a process known as "the glymphatic system" [13].

Mechanistically, the inflow of CSF into the brain parenchyma during sleep is believed to be driven by an interplay of cerebral vasomotion and the respiratory and cardiac cycles. Cerebral vasomotion, referring to low-frequency constrictions and dilations of the cerebral vasculature, facilitates CSF flow along periarterial spaces and into the brain parenchyma [14,15]. In parallel, respiratory in- and exhalations [16–18] together with the heartbeat [17,19,20] generate pressure waves that propagate through the brain, physically propelling CSF into and through CSF spaces and the brain [16–20]. Recent discoveries suggest that vasomotion itself is regulated by neural activity, norepinephrine, and the sleep–wake cycle [15,21–23]. This critical role of physiological drivers for CSF flow is underscored by pharmacological studies, demonstrating that enhancing arterial pulsatility via systemically administered adrenergic agonist directly affects both CSF flow within perivascular spaces and CSF-interstitial fluid exchange throughout the mouse brain [20,24].

Accumulating evidence supports the role of sleep in facilitating brain fluid flow and waste clearance in humans. A night of sleep is associated with increases in brain diffusivity, CSF production, and CSF volume [25–27], while light NREM sleep enhances pulsatile movements of CSF in the fourth ventricle [28]. Consistent with this, diffusivity in white matter (WM) regions of the brain decreases from morning to evening [29],

and sleep deprivation results in reduced clearance of contrast agent from the brain [30] as well as increased accumulation of tau and β-amyloid in the CSF and brain parenchyma [31–33]. Further aligning with findings from rodent studies, invasively measured intracranial pressure (ICP) B-waves (0.5–2 waves pr min), thought to reflect cerebral vasomotion [34], are also influenced by the sleep–wake cycle and sleep stage [35,36].

However, understanding how changes in vigilance state and NREM slow waves relate to CSF flow in the healthy human brain requires noninvasive methods that allow for natural sleep during continuous measurements of both brain fluid dynamics and EEG. Accelerated brain imaging techniques, such as Magnetic Resonance Encephalography (MREG) [37], have emerged as valuable tools for this purpose. MREG is a functional T2*-weighted imaging method with millisecond-level temporal resolution, enabling noninvasive assessment of low-frequency brain pulsations (LFPs) as well as brain pulsations in respiratory and cardiac cycle frequency ranges. LFPs are thought to reflect vasomotion, while the strength of respiration- and cardiac-driven brain pulsations are suggested to reflect both vascular and extravascular factors, such as CSF and water content in the brain [17,38]. Accordingly, variability in these pulsations may indicate changes in brain fluid flow. Previous human studies have shown that NREM sleep stages N1 and N2 enhance all three types of brain pulsations [28,39,40], with sleep effects involving larger brain regions in N2 than in N1 sleep [40]. Additionally, LFPs increase during drowsiness [41], in N1 sleep following sleep deprivation [40], and with autonomic arousal [42]. Although the influence of circadian rhythm and sleep pressure on these findings remains unclear, they suggest that both NREM sleep, sleep depth and wake-related factors modulate LFPs and respiration- and cardiac-driven brain pulsations, with potentially distinct effects on the three.

Our study aimed to establish how the strength of MREG-detected brain pulsations correlates with heightened sleep pressure, slow-wave-rich NREM sleep, sleep depth, and EEG delta power, while controlling for circadian rhythm, time of day, and caffeine intake. To do this, we assessed the association between simultaneously acquired MREG and EEG signals in healthy individuals during well-rested wakefulness, after 35 h of prolonged wakefulness, and during NREM sleep stages N2 and N3. Moreover, we examined the effect of pharmacologically induced vasodilation using the α1- and β-adrenergic antagonist carvedilol (Fig 1).

We hypothesized that (1) sleep deprivation is associated with a homeostatic drive towards enhanced brain pulsations in wakefulness, (2) NREM sleep further enhances these pulsations, (3) the strength of brain pulsations is proportional to sleep depth and EEG delta power, and (4) adrenergic antagonism modulates brain pulsations.

## Results

### Study cohort and descriptives

The study included 20 healthy males with anamnestic and polysomnographically verified normal sleep (Table 1; Fig 2). Three MR/EEG scans sessions were conducted after an average of 11 and 35 h of wakefulness (standardized time: ~18.30–20.30; Fig 1; S1 Table). Participants adhered to their sleep–wake schedules, verified by actigraphy and sleep diary, ensuring that all scans were performed at same circadian time point. Wakefulness during sleep deprivation was confirmed with continuous EEG recordings (S1 Table). Sleep patterns were comparable across the three standardized nights leading up to scan days (S2 Table). During well-rested scans, participants were mostly awake, but once sleep-deprived, they quickly entered NREM sleep stages N2 and N3 (S1 Fig).

Increased sleep pressure due to sleep deprivation was confirmed by higher awake EEG delta power in sleep-deprived compared to well-rested scans (delta power ratio: +0.033 ± 0.017 AU, $p = 0.051$, $N = 20$) and impaired psychomotor vigilance performance before scan sessions (reaction time: −0.035 ± 0.005 s; lapses of attention +3.86 ± 0.23; $p_{both} < 0.001$, $N = 20$). When sleep deprived, plasma norepinephrine levels (measured after scan sessions) increased numerically (+268.8 ± 114.5 pg/mL, $p_{adj} = 0.074$, $N = 20$, S2 Fig).

As expected, NREM sleep was associated with a strong increase in EEG delta power (+0.081 ± 0.014 AU, $p < 0.001$, $N = 20$) compared to sleep-deprived wakefulness, with N3 exhibiting higher EEG delta power than N2 sleep ($p < 0.001$,

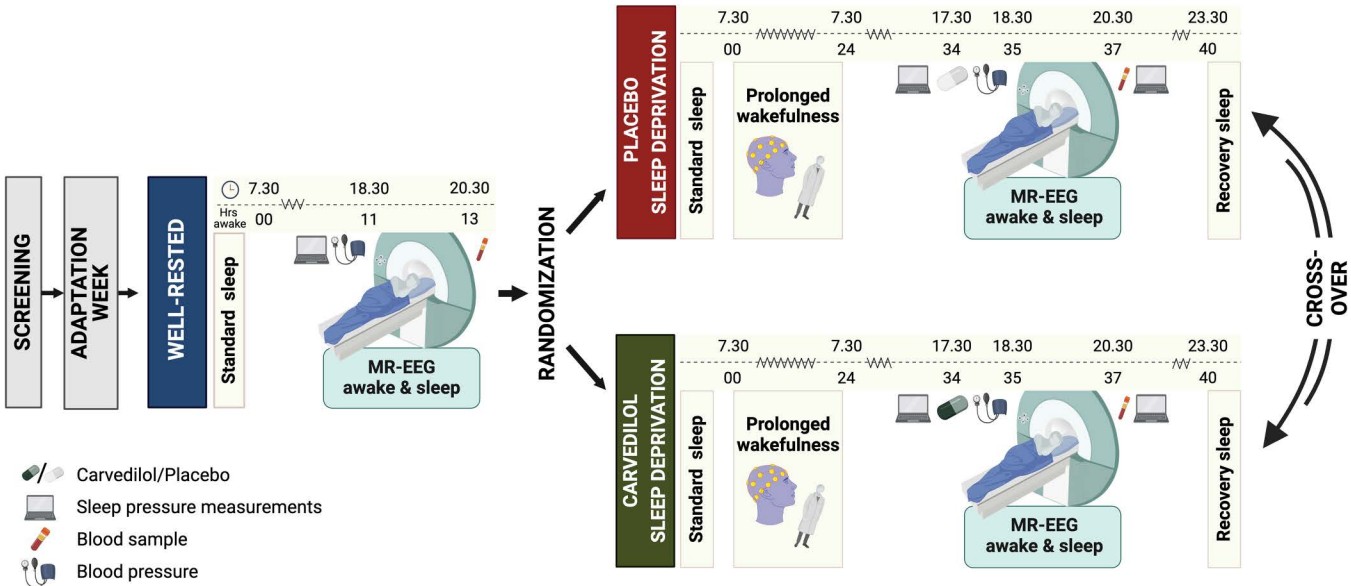

**Fig 1. Study design of the sleep study.** A well-rested and two sleep-deprived study sessions were conducted in the evenings at similar circadian time points. All three MR/EEG scans included a period of wakefulness followed by a sleep opportunity. Before the two sleep-deprived scans, participants received either a placebo or 25 mg of the adrenergic antagonist carvedilol in a double-blind, randomized, cross-over manner. N = 20. *Created in BioRender. Rigshospitalet, N. (2025)* https://BioRender.com/.w2bbiai.

**Table 1. Demographics.**

|  | Mean | SD | min–max |
| --- | --- | --- | --- |
| **Age (years)** | 24.1 | 2.8 | 20–29 |
| **Body mass index (kg/m²)** | 22.7 | 2.8 | 19.1–31.6 |
| **Reported habitual sleep duration (h)** | 7.6 | 0.4 | 7.0–8.0 |
| **Morningness-eveningness (MEQ)** | 48.6 | 6.4 | 38.9–61.0 |
| **General daytime sleepiness (ESS)** | 6.2 | 3.0 | 2.0–13 |
| **Sleep quality index (PSQI)** | 3.0 | 1.6 | 1.0–6.0 |

Demographic characteristics of the 20 healthy males included in the study. MEQ, Morningness-eveningness questionnaire; PSQI, Pittsburgh Sleep Quality Index; ESS, Epworth Sleepiness Scale.

$N = 17$). NREM sleep also showed a small drop in respiration rate (−9%, $p = 0.01$) and heart rate (−11%, $p < 0.001$) (S3 Table), while the heart rate increased slightly during N3 versus N2 sleep (1.6%, $p = 0.01$) with no change in respiration rate (S5 Table). Treatment blinding was successful; participants guessed whether they received carvedilol or placebo before the two sleep-deprived scans only at chance level (guess accuracy: 54%).

## Sleep deprivation enhances spectral power in LFP frequency band

Effects of sleep deprivation and NREM sleep on whole-brain LFPs (0.012–0.034 Hz; 0.72–2 waves pr. min) were evaluated in 5-min MREG scans from well-rested and sleep-deprived placebo scan sessions (S4 Table; *see* Materials and methods).

For the 5-min wakefulness scans included in analysis, an average of 98% of the simultaneously recorded EEG was classified as wakefulness, with none of the remaining 2% consistently classified as either N1, N2, or N3 sleep. For sleep

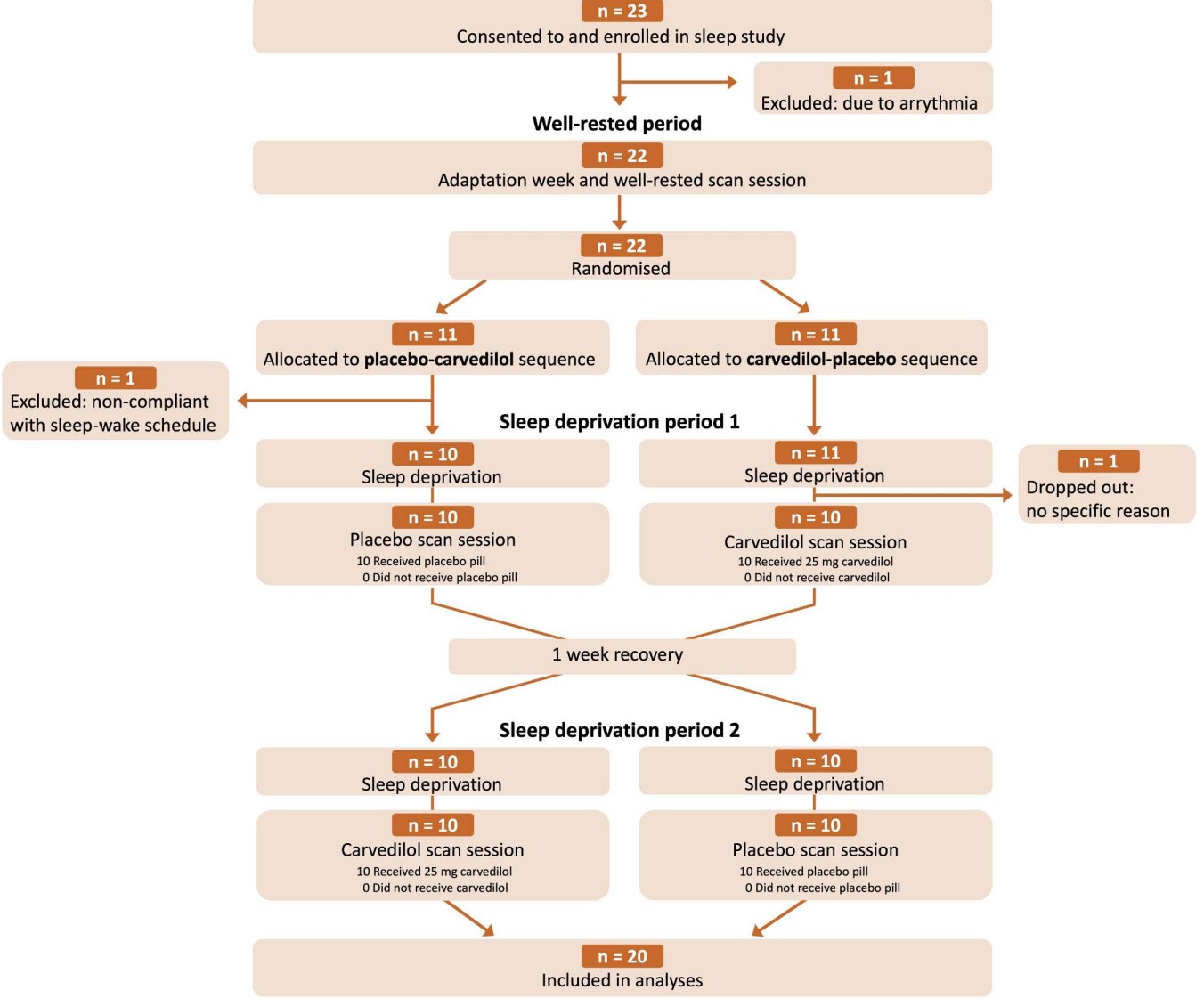

**Fig 2. Consort flow diagram.** Number of research volunteers enrolled, excluded, allocated to intervention sequence, and who completed the study and were included in analyses.

scans, 91% was classified as N2 or N3 sleep, 0.02% as wakefulness, and the remainder as N1 sleep, artifacts, or scorer disagreement.

Sleep deprivation was associated with a 120% increase in awake spectral power within the LFP frequency band (estimated mean in sleep-deprived ($10^{7.091}$) versus well-rested wakefulness ($10^{6.748}$), $120\% = (10^{7.091} - 10^{6.748})/10^{6.748}$, $p = 0.030$; Fig 3B). During sleep-deprived NREM sleep (stages N2 and N3), LFP spectral power was in-between that of rested wakefulness and of sleep-deprived wakefulness, with no significant effect of sleep observed ($p = 0.19$; Fig 3A–3C).

To investigate the relationship between sleep pressure and LFPs, we next examined the association between LFP spectral power and psychomotor vigilance, as quantified with the psychomotor vigilance test (PVT). We observed a correlation between LFP spectral power during NREM sleep and pre-scan PVT performance (lapses of attention: $r_{adj} = 0.50$,

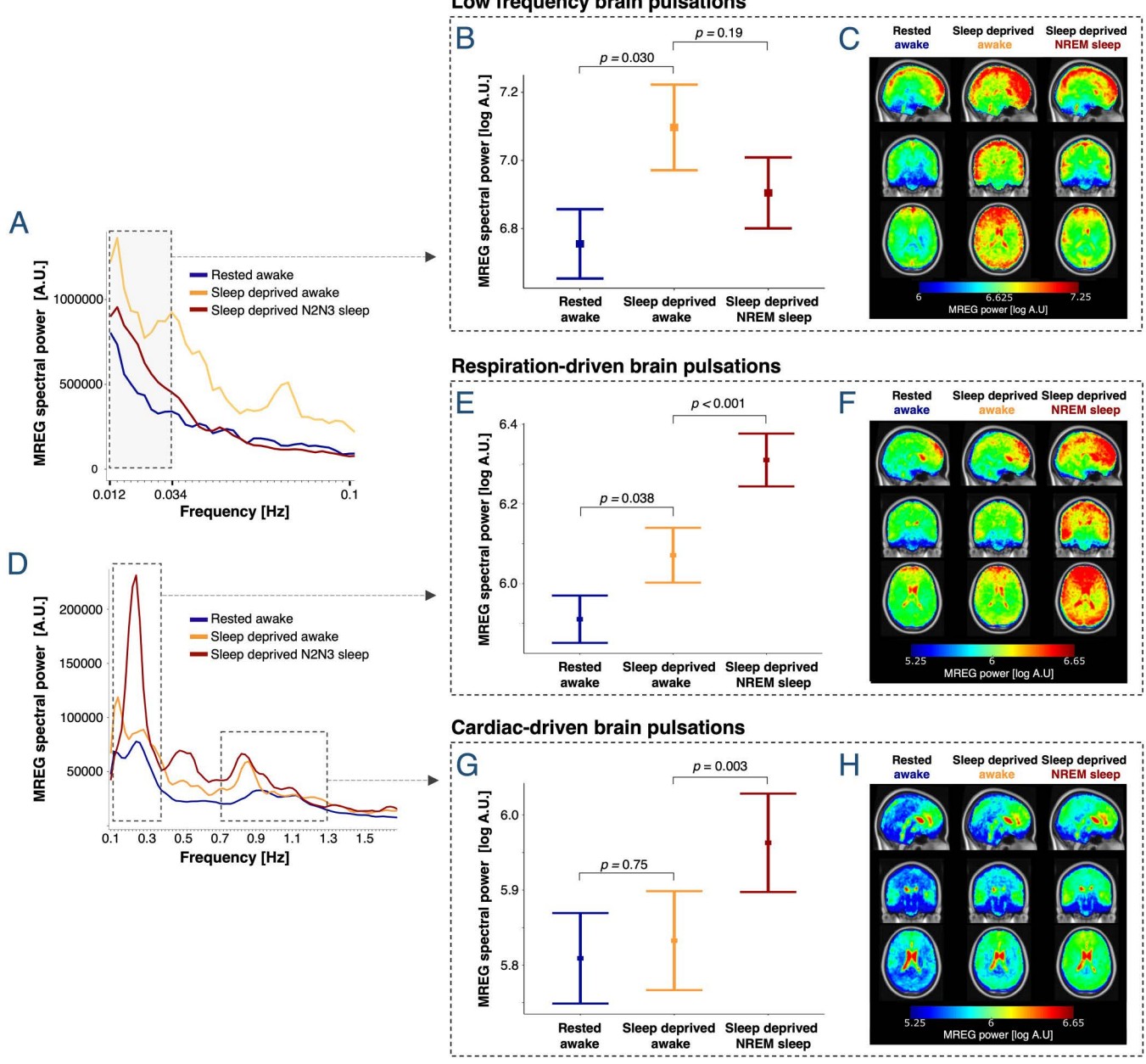

**Fig 3. Sleep deprivation increases LFP strength, while NREM sleep is the primary enhancer of respiration- and cardiac-driven brain pulsations. (A**, **D)** Mean whole-brain MREG spectra across participants. Dashed boxes denote frequency ranges for LFPs (0.012–0.034 Hz) and for respiration- and cardiac-driven brain pulsations. (**B, E, G)** Estimated means ± SEM and *p*-values are from linear mixed models evaluating effects of sleep deprivation and NREM sleep on log spectral power. (**C, F, H)** Whole-brain maps illustrate the regional spectral power [log A.U.] in the LFP, respiration, and cardiac frequency bands across all 5-min scans (LFP) or 30-s epochs (respiration and cardiac), averaged across first epochs/scans, then participants (warped into MNI space). *N* included in analyses = 20 (see S3 and S4 Tables for number of participants in the three conditions).

*p* = 0.018; reaction time: $r_{adj}$ = 0.49, *p* = 0.023; Fig 4A and 4B). Improvement in PVT performance after a nap in the scanner (i.e., shorter reaction time and fewer lapses) also correlated positively with strength of LFP spectral power during NREM sleep (lapses: $r_{adj}$ = −0.48, *p* = 0.015; reaction time: $r_{adj}$ = −0.39, *p* = 0.043; Fig 4C and 4D). We observed no such correlation during either rested or sleep-deprived wakefulness ($p_{all}$ > 0.5).

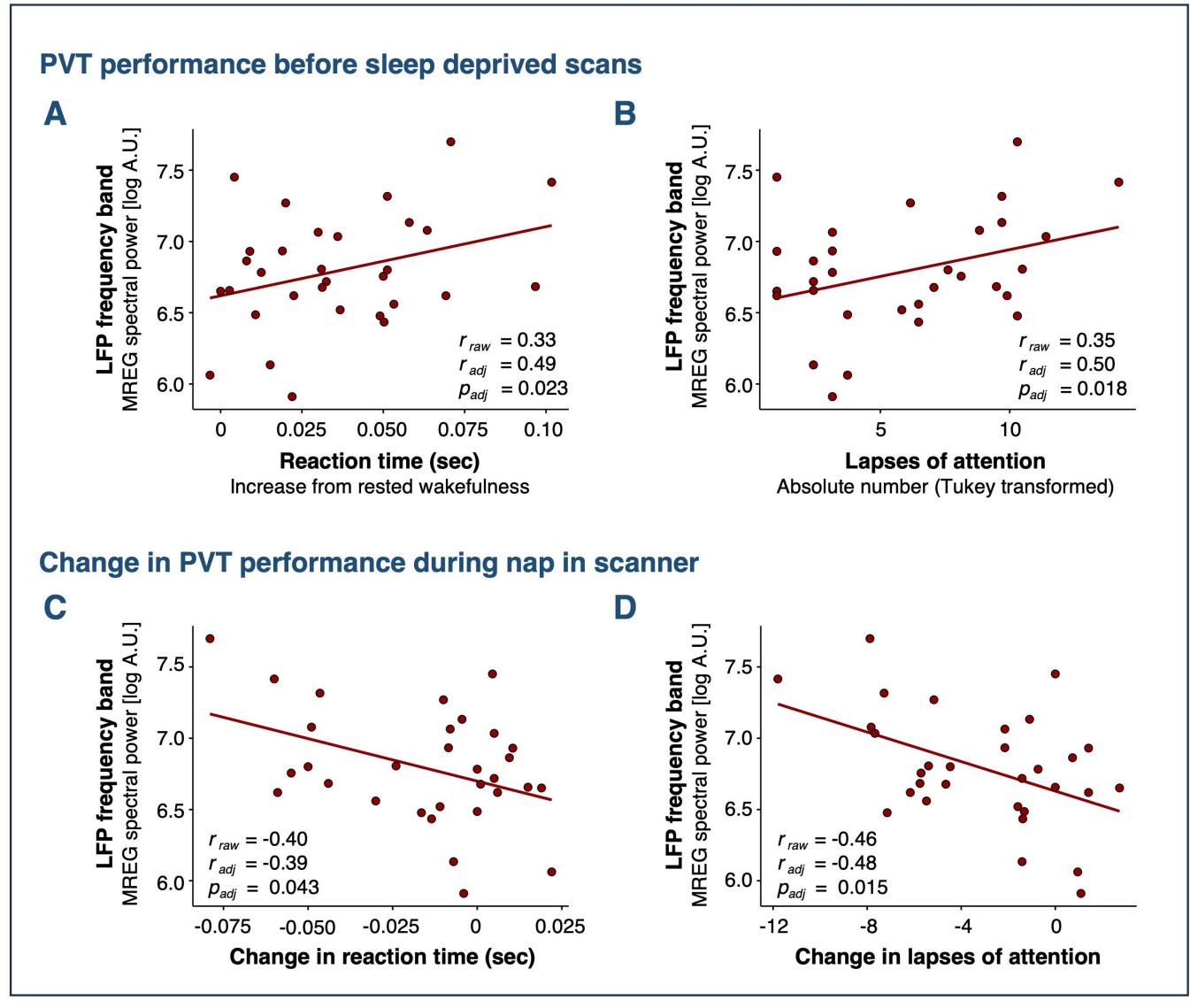

**Fig 4. PVT performance is associated with LFPs during sleep. (A, B)** Correlations between pre-scan measures of impaired PVT performance (i.e., longer reaction times and more lapses of attention) and intensity of LFP spectral power during sleep. **(C, D)** Improvement in PVT performance (i.e., lower reaction time and less lapses) from before to after nap in scanner correlates with strength of LFP spectral power during sleep. Data from both sleep-deprived scans are included. Dots represent mean LFP spectral power during NREM sleep (stages N2 or N3). $r_{raw}$ denotes Pearson's corelation coefficient. $r_{adj}$ and $p_{adj}$ denote estimates from linear mixed models adjusted for treatment and repeated measures. $N_{placebo} = 14$, $N_{carvedilol} = 17$.

A sensitivity analysis based on 2-min periods fully classified as either wakefulness or NREM sleep, rather than the 5-min scans, confirmed our main finding. It showed a 96% increase in awake LFPs following sleep deprivation ($p = 0.038$), with no significant effects of NREM sleep (see supplementary materials for details; S1 File).

**NREM sleep enhances spectral power in the respiration and cardiac frequency bands; N3 more so than N2 sleep**

The effects of sleep deprivation, NREM sleep, and sleep depth on brain pulsations driven by respiratory and cardiac forces were evaluated in 30-s epochs from well-rested and sleep-deprived placebo scans, in which our two independent EEG-scorers agreed on sleep staging (S3 and S5 Tables). MREG spectral power within the respiration and cardiac

frequency bands was individually tailored for each epoch based on simultaneously recorded respiration and heart rate signals, which were also included in the statistical models to account for changes in cardiorespiratory rates (see Materials and methods).

Compared to rested wakefulness, awake sleep-deprived participants had increased whole-brain spectral power within the respiration frequency band (+45%, $p = 0.038$, Fig 3D, 3E, and 3F), but not in the cardiac frequency band (+6%, $p = 0.75$, Fig 3D, 3G, and 3H). When participants fell asleep and entered NREM sleep (pooled N2 and N3), we observed a strong increase in whole-brain spectral power in both the respiration (+73%, $p < 0.001$, Fig 3D, 3E, and 3F) and cardiac frequency bands (+35%, $p = 0.003$, Fig 3D, 3G, and 3H). Compared to rested wakefulness, spectral power in sleep-deprived NREM sleep increased by 150% and 42% in the respiration and cardiac frequency bands, respectively ($p < 0.001$).

Sleep depth also influenced respiration- and cardiac-driven brain pulsations. In the respiration frequency band, N2 sleep was associated with a 35% increase in spectral power compared to sleep-deprived wakefulness ($p_{adj} = 0.047$), while N3 sleep further increased spectral power by 57% ($p_{adj} < 0.001$) (Fig 5A, 5B, and 5C). In the cardiac frequency band, spectral power was higher during N3 sleep than during both N2 sleep (+39%, $p_{adj} < 0.001$) and sleep-deprived wakefulness (+58%, $p_{adj} < 0.001$). No statistically significant change in cardiac spectral power was observed when comparing sleep-deprived wakefulness to N2 sleep ($p_{adj} = 0.19$) (Fig 5A, 5D, and 5E). We did not observe any correlation between PVT performance and spectral power in the respiration and cardiac frequency bands ($p_{adj, all} > 0.18$).

Lastly, to confirm that effects of vigilance state on brain pulsations in cardiorespiratory frequency ranges are indeed linked to these physiological processes, we assessed MREG spectral power in a control frequency band (3.54–3.74 Hz), matched in width to the cardiac band and located above typical heart rates and their harmonics. This analysis revealed no significant effect of sleep deprivation ($p = 0.685$) or NREM sleep stages N2 and N3 ($p = 0.257$).

### EEG delta-power during MR imaging correlates with spectral power in respiration and cardiac frequency bands

To validate the found effects of NREM sleep and sleep depth on respiration- and cardiac-driven brain pulsations in a quantitative and sleep-scoring independent manner, we quantified EEG delta-power ratio across all 30-s epochs from well-rested and sleep-deprived placebo scans, irrespective of their vigilance state classification (Fig 5F and 5G). This showed a positive correlation between EEG delta power ratio and whole-brain spectral power in both the respiration ($r_{raw} = 0.39$, $r_{adj} = 0.27$, $p_{adj} < 0.001$; Fig 5F) and cardiac ($r_{raw} = 0.26$, $r_{adj} = 0.19$, $p_{adj} < 0.001$; Fig 5G) frequency bands. This positive association was identifiable in most participants (S3 Fig).

### NREM sleep primarily enhances spectral power in respiration and cardiac frequency bands in gray and white matter

To explore whether the strength of physiological brain pulsations differs between gray matter (GM), WM, and CSF, we analyzed data from well-rested and sleep-deprived placebo scans. During rested wakefulness, LFP spectral power was consistent across tissue types (Fig 6A). In sleep deprivation, LFP spectral power also increased to a similar extent in GM, WM, and CSF ("tissue type" × "sleep deprivation" interaction: $p = 0.96$; S4A Fig). Similarly, no interactions were observed between NREM sleep and tissue type for LFPs ($p = 0.92$; Fig 6B and 6C).

By contrast, in the respiration and cardiac frequency bands, spectral power varied significantly across CSF, GM, and WM ("tissue type" main effect: $p < 0.001$). During rested wakefulness, CSF showed notably higher spectral power than both GM (respiration band: +101%, $p_{adj} < 0.001$; cardiac band: +1,030%, $p_{adj} < 0.001$) and WM (respiration band: +118% $p_{adj} < 0.001$; cardiac band: +1,153%, $p_{adj} < 0.001$; Fig 6A). Additionally, participants had higher spectral power in GM than in WM in the well-rested condition (respiration band: +9%, $p = 0.017$, $p_{adj} = 0.052$; cardiac band: +11%, $p = 0.005$, $p_{adj} = 0.014$; Fig 6A).

The effect of NREM sleep on respiration- and cardiac-driven brain pulsations was more pronounced in GM and WM than in CSF, with a significant interaction effect between vigilance state and tissue type ($p < 0.001$; Fig 6B and 6C).

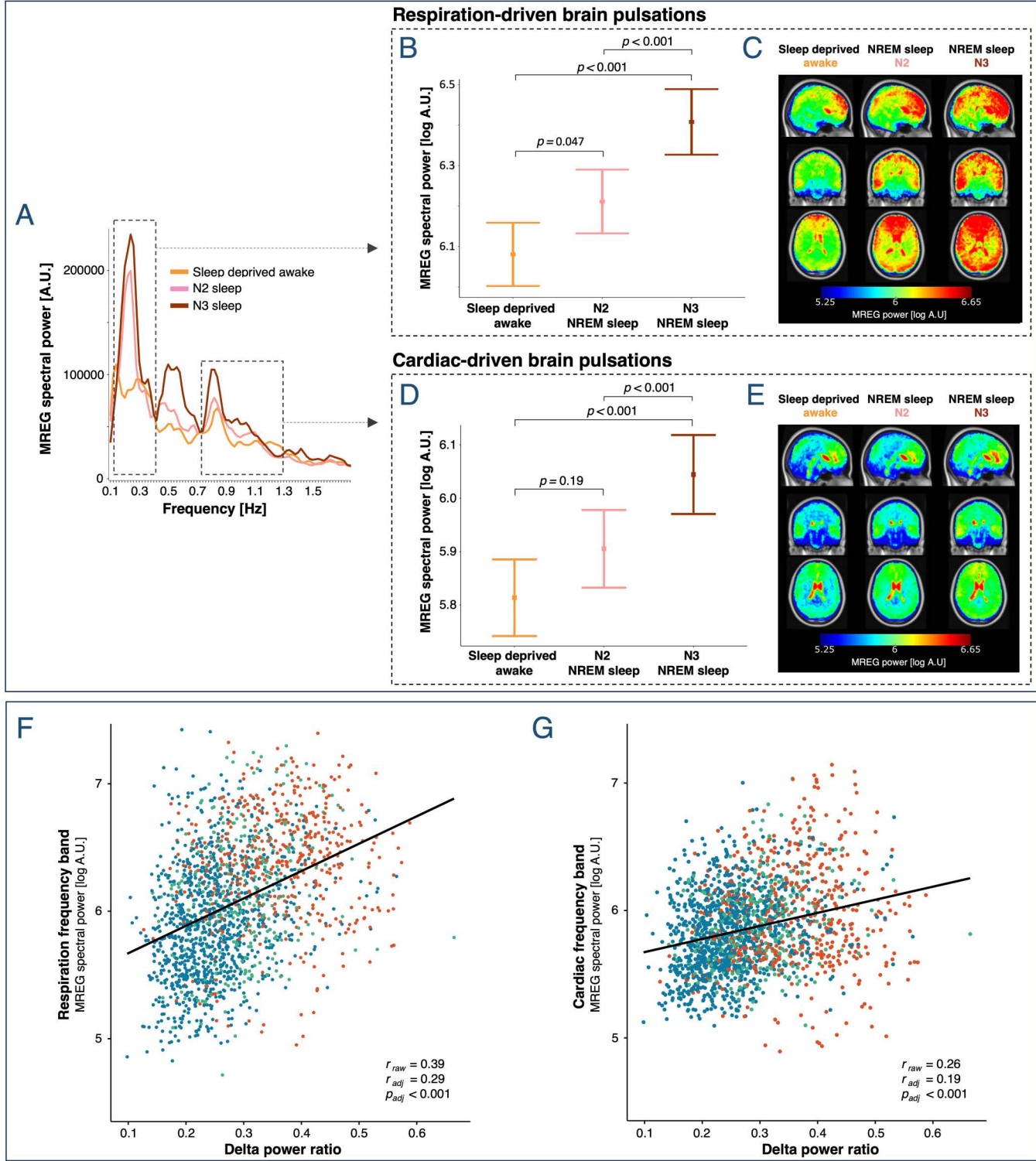

**Fig 5. Strength of respiration- and cardiac-driven brain pulsations correlates with sleep depth and EEG delta power. (A)** Mean whole-brain MREG spectra across participants for sleep-deprived wakefulness, N2 sleep, and N3 sleep (placebo scans). Dashed boxes denote frequency ranges, in which respiration- and cardiac-related peaks are found. **(B, D)** Estimated means ± SEM and *p*-values from linear mixed models evaluating the effect of sleep depth on log spectral power. All *p*-values have been adjusted for multiple comparison. **(C, E)** Whole-brain maps illustrate the regional spectral

power [log A.U.] in the respiration and cardiac frequency bands across all 30-s epochs, averaged across first epochs, then participants (warped into MNI space). $N = 19$ (see S5 Table for number of participants in each of the conditions). **(F,G)** EEG delta power ratio vs. simultaneously acquired spectral power in the respiration and cardiac frequency bands in all 30-s epochs from both rested and sleep-deprived placebo scans. Colors represent the scored sleep stages (blue: wakefulness; red: NREM sleep stage 2 and 3; green: uncertain scorings) and are included for visualization only. Sleep stage was not included in analysis. $r_{raw}$ denotes Pearson's correlation coefficient. $r_{adj}$ and $p_{adj}$ denote adjusted estimates from linear mixed models. $N_{resp} = 19$ and $N_{card} = 20$.

Specifically, GM and WM exhibited 70% and 75% stronger increases in spectral power during NREM sleep than CSF in the respiration frequency band, and 92% and 110% stronger increases than CSF in the cardiac frequency band ($p_{adj, all} < 0.001$). When evaluated separately, all tissue types showed significant effects of NREM sleep on spectral power in both respiration- and cardiac frequency bands (respiration band$_{(all)}$: $p < 0.001$; cardiac band$_{(all)}$: $p < 0.01$). Additionally, sleep deprivation interacted with tissue type in the respiration band ($p < 0.001$), with effects confined to GM ($p = 0.045$) and WM ($p = 0.054$) (S4B and S4C Fig).

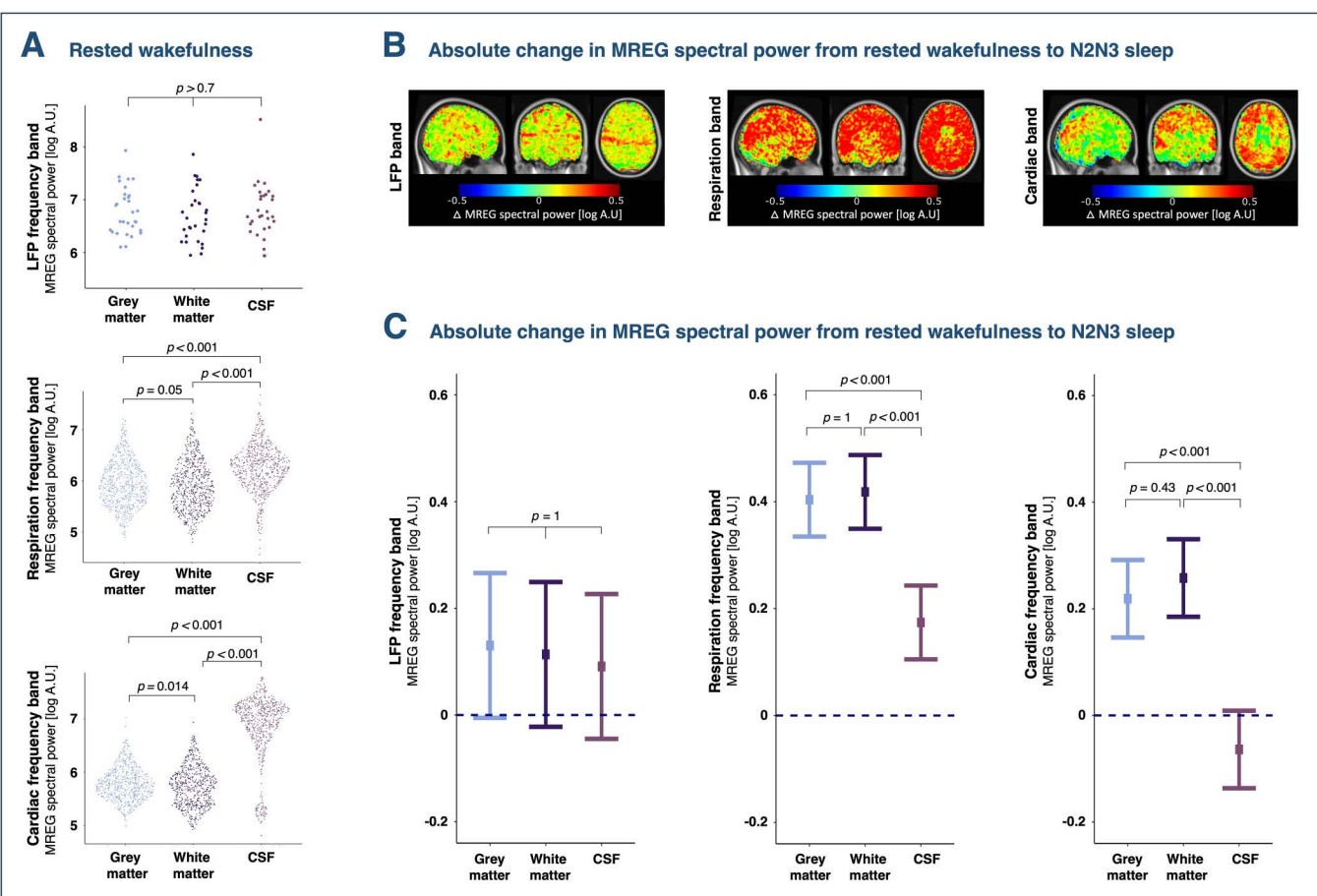

**Fig 6. Sleep has a greater effect on respiration- and cardiac-driven brain pulsations in gray and white matter than in CSF. (A)** Difference in brain pulsation strength between gray matter, white matter, and CSF in ventricles in rested wakefulness. Each data point represents a 30-s epoch from a single participant. **(B)** Brain maps illustrate the averages of participant-wise absolute differences in spectral power between rested wakefulness and NREM sleep (stages N2 and N3) in LFP, respiration, and cardiac frequency bands. Whole-brain spectral power maps of individually calculated voxel-wise differences were warped into MNI space and averaged across participants. **(C)** Changes in estimated means of log spectral from rested wakefulness to N2–N3 NREM sleep. $p$-values represent interaction effects between tissue type and sleep from linear mixed models and are adjusted for multiple comparisons. $N = 20$ (see S3 and S4 Tables for number of participants in each condition).

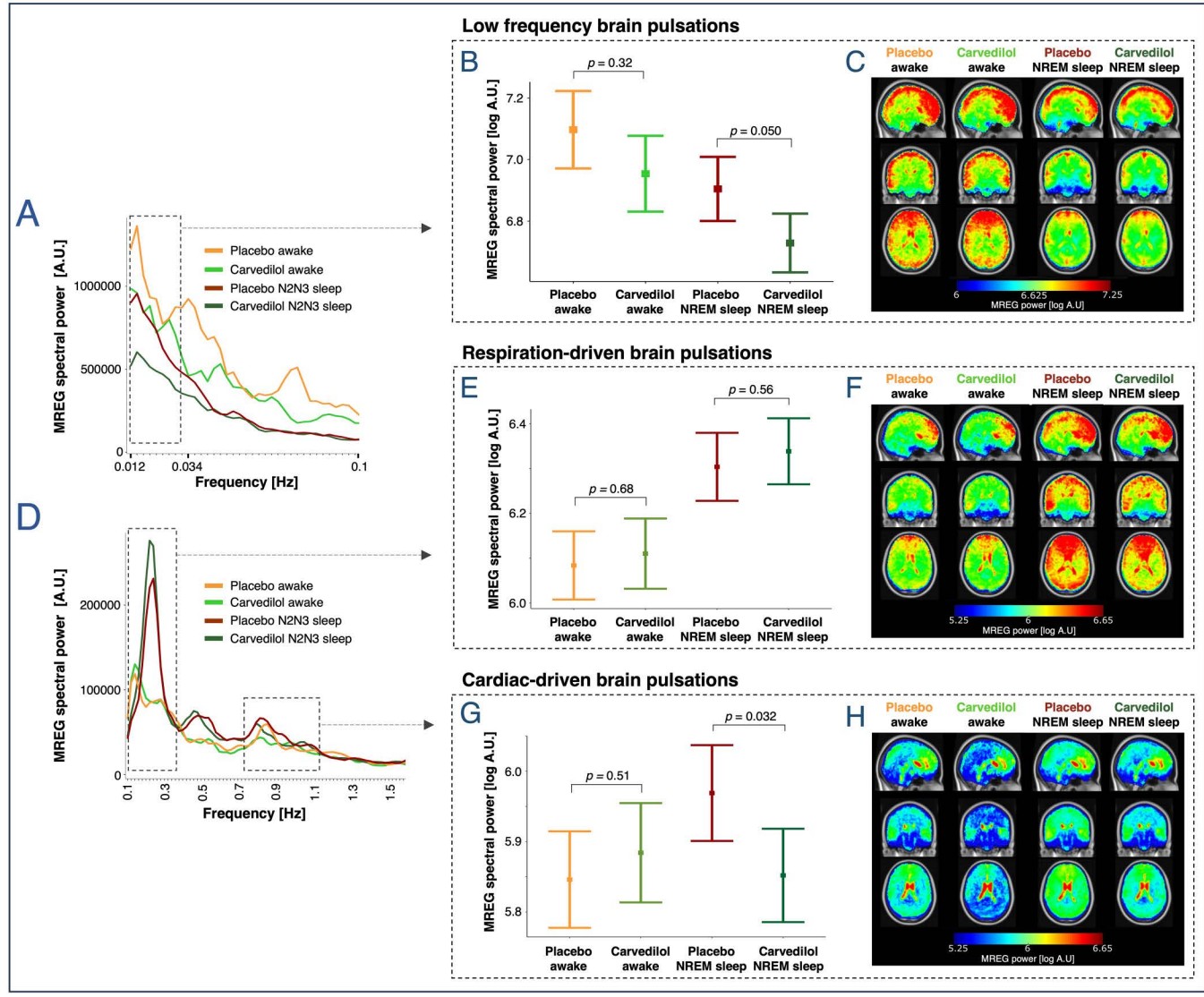

**Fig 7. The α1 and β-adrenergic antagonist carvedilol attenuates LFPs and cardiac-driven brain pulsations during sleep. (A, D)** Mean whole-brain MREG spectra across participants in the sleep-deprived scans for placebo wakefulness, placebo NREM sleep, carvedilol wakefulness, and carvedilol NREM sleep. Dashed boxes denote frequency ranges for LFPs (0.012–0034 Hz) and for respiration- and cardiac-related peaks). **(B, E, G)** Estimated means ± SEM and *p*-values are from linear mixed models evaluating the effect of treatment and NREM sleep. **(C, F, H)** Brain maps illustrate regional MREG spectral power [log A.U.] in the LFP, respiration, and cardiac frequency bands across participants, averaged across first all 5-min scans (LFP) or 30-s epochs (respiration and cardiac) per participant, then across participants (warped into MNI space). *N* = 20 (see S3 and S4 Tables for number of participants in each of the three conditions).

## Adrenergic antagonism decreases spectral power in LFP and cardiac frequency bands

Treatment with the adrenergic antagonist carvedilol demonstrated the anticipated systemic effects: It reduced plasma norepinephrine levels [43] (−479 ± 113 pg/mL, $p_{adj} < 0.001$, *N* = 20; S2 Fig), lowered mean arterial blood pressure by ~5% (−4.4 ± 0.9 mmHg, *p* < 0.001, *N* = 20), and did not alter respiration rates or heart rates (S3 Table). Carvedilol also did not change absolute or relative time spent in wakefulness or sleep during scans (S1 Fig) and only had minor effects on recovery sleep (S6 Table). During NREM sleep, carvedilol was associated with a decrease in LFP spectral power (−51%, *p* = 0.050), while this was not the case during wakefulness (−35%, *p* = 0.32) (Fig 7A—7C). Similarly, carvedilol significantly

reduced spectral power in the cardiac frequency band during NREM sleep (−30%, $p = 0.032$), but not during wakefulness (Fig 7D, 7G, and 7H). In the respiration frequency band, carvedilol intervention showed comparable spectral power outcome to placebo during both sleep-deprived wakefulness and NREM sleep ("treatment": $p = 0.56$, "NREMsleep$_{carvedilol}$": $p < 0.001$, Fig 7D, 7E, and 7F).

These carvedilol-versus-placebo estimates were unaffected by whether scan-day (period effect) was modeled as a random intercept or an explicit fixed effect (see Materials and methods).

## Discussion

In this circadian-controlled sleep and sleep deprivation study, we demonstrate that heightened sleep pressure promotes low frequency cerebral vasomotion (0.012–0.034 Hz; 0.07–2 waves pr min), while brain pulsations driven by the respiratory and cardiac cycles intensify during NREM sleep stages N2 and N3, correlating with both sleep depth and EEG delta power (see visual representation of results in Fig 8).

The strength of LFPs and respiration- and cardiac-driven brain pulsations, as measured with T2*-weighted accelerated imaging techniques, has been suggested to reflect either the magnitude of CSF flow in perivascular spaces and the brain parenchyma [17,38] or the underlying compliance of brain tissue, encompassing factors such as brain CSF and water amount and tissue viscosity [44,45]. Our data support this interpretation for respiration- and cardiac-driven brain pulsations demonstrating that, irrespective of vigilance state, these pulsations are significantly stronger in the CSF-filled ventricles than in GM and WM. During rested wakefulness, they are also more pronounced in GM compared to WM. Considering the differences in water and CSF content, as well as tissue viscosity, across these three brain regions [44,45], our results indicate that the relative water and CSF content of a given region (or voxel) is closely associated with the strength of the MREG-detected respiration- and cardiac-driven brain pulsations. Together, this suggests that an increase in CSF influx into perivascular spaces and the brain parenchyma will amplify brain pulsatility. For LFPs, we observed no differences in pulsation strength across tissues types, supporting the view that MR-detected brain pulsations below 0.1 Hz reflect global cerebral vasomotion and its temporally associated effects on CSF pulsation [28,42]. Notably, we discovered that intervention with the adrenergic antagonist carvedilol—which is known to cause peripheral vasodilation through inhibition of smooth muscle contraction [46,47]—was associated with a reduction in LFP strength. This observation supports

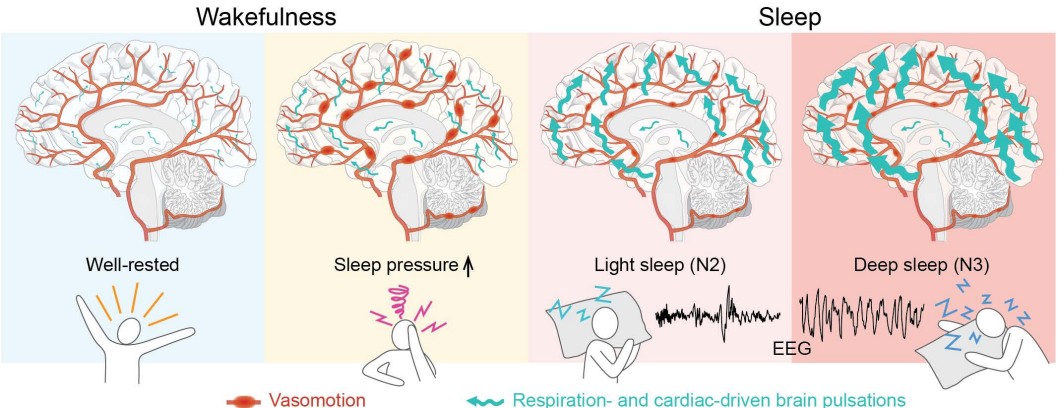

**Fig 8. Sleep pressure and slow-wave-rich NREM sleep exert distinct effects on cerebral vasomotion and on brain pulsations driven by respiratory and cardiac forces.** Heightened sleep pressure (following 35 h of wakefulness) promotes vasomotion (yellow box). Slow-wave-rich NREM sleep enhances brain pulsations driven by the respiratory and cardiac cycles—more so in gray and white matter than in the ventricles (red boxes). The respiration- and cardiac-driven brain pulsations also intensify with deeper sleep (N3 > N2) and correlate with EEG delta power, which is a measure of slow-wave activity (light vs. dark red boxes).

the current understanding that LFPs originate from cerebral vasomotion, and offers encouraging evidence that vasomotion may be amenable to pharmacological modulation.

A novel finding of our study is that sleep deprivation has a greater effect on LFPs than sleep itself. Specifically, MREG-detected LFPs increase at a whole-brain level after 35 h of wakefulness, compared to both rested wakefulness and subsequent periods of NREM sleep stages N2 and N3. Additionally, LFP strength during sleep correlates with pre-sleep measures of impaired psychomotor vigilance, which is a well-established marker of homeostatic sleep pressure, or the biological need to sleep [48,49]. In line with this, we also found a positive correlation between LFP strength during sleep and improvements in psychomotor vigilance from before to after the nap. Collectively, this suggests sleep pressure as a strong driver of vasomotion and that these low-frequency constriction-dilation dynamics of the cerebral vasculature may contribute to performance restoration during sleep. Supporting this notion, a recent MREG study showed that healthy individuals exhibit stronger LFPs during sleep-deprived N1 sleep than during rested N1 sleep [40]. While previous studies in humans have demonstrated that NREM sleep stages N1 and N2 increase LFP strength [28,40,39], they did not differentiate the effects of sleep deprivation from those of sleep itself, nor did they account for circadian influences or sleep pressure at scan onset. In the present study, we address these gaps, demonstrating that LFPs are already elevated during sleep-deprived wakefulness and therefore, we propose that earlier reports of LFPs in sleep have been confounded by sleep pressure effects. Sleep stage might also influence LFPs. ICP B-waves (0.5–2 waves pr min), which are believed to reflect cerebral vasomotion [34], have been reported to occur more frequently in N1 and N2 sleep [35,36] and less so during deeper sleep [50]. Similarly, a study of nonsleep-deprived individuals showed that both the LFP signal of CFS in the fourth ventricle and its simultaneous global signal are strongest in N1 sleep, followed by N2 sleep, and weakest in N3 sleep [42]. Given that sleep pressure dissipates over time spent asleep [3], this indicates that vasomotion is strongest at the beginning of sleep, when sleep pressure is at its peak and individuals typically reside in lighter sleep stages. Lastly, because autonomic arousal has been proposed as a potential mechanism for LFP regulation [42], we examined markers of sympathetic activity between well-rested and sleep-deprived wakefulness. However, in our data, blood pressure, heart rate, and respiration rate did not differ between the two conditions, indicating that factors beyond both neural slow waves and autonomic arousal likely contribute to the increase in awake LFPs following sleep deprivation. Taken together, these findings indicate that cerebral vasomotion in the 0.012–0.34 Hz range (0.07–0.2 waves pr min) intensifies as the need for sleep increases. This is especially compelling because low-frequency vasomotion is believed to be a key driver of CSF influx to the brain parenchyma [14]. Based on this, we hypothesize that as sleep pressure builds up, the brain activates restorative brain clearance mechanisms.

Expanding on previous findings [40,39], we show that slow-wave-rich NREM sleep (pooled N2 and N3 sleep) enhances respiration- and cardiac-driven brain pulsations at a global brain level, and as a novel finding, we show that sleep-effects are more pronounced in GM and WM than in the CSF-filled ventricles. Because the strength of these MREG-detected brain pulsations likely scales with brain water and CSF content, their enhancement during NREM sleep is compatible with a sleep-dependent increase in CSF influx into perivascular spaces and the brain parenchyma accompanied with an expansion of extracellular space volume. This is consistent with observations made in the mouse brain [8,10]. Notably, when participants fell asleep and entered NREM sleep, the strength of respiration- and cardiac-driven brain pulsations increased by 73% and 35%, which can cautiously be compared to the ~60% increase in interstitial space volume reported by Xie and colleagues in naturally sleeping rodents [8].

We further demonstrate that the strength of respiration- and cardiac-driven brain pulsations increases with the sleep depth, showing greater pulsatility in N3 than in N2 sleep, and correlates positively with EEG delta power, which is a measure of slow wave activity. A recent study on shallower NREM sleep [40] agrees with these results, reporting that respiration- and cardiac-driven brain pulsations encompass larger brain regions in N2 than in N1 sleep. Our data cannot determine how CSF influx to the brain increases with sleep intensity, as we did not observe significant increases in cerebral vasomotion by sleep. However, the association between more sleep slow waves—i.e., deeper sleep and higher delta power—and the increase in strength of respiration- and cardiac-driven brain pulsations is in line with the recent suggestion

that the synchronous neuronal activity of slow-wave sleep may physio-mechanically drive fluid and solute movement into and through the brain [51,52]. As EEG delta power is also an established marker of homeostatic sleep propensity [2,53], its close association with respiration- and cardiac-driven brain pulsation strength, and thus brain fluid flow, implies that this may be homeostatically regulated and an integral function of human sleep. Our results closely mirror preclinical work demonstrating that CSF influx and perfusion through the brain parenchyma is associated with EEG slow waves [9,51]. Consequently, we here provide support for the hypothesis that humans—alike mice – have a sleep- and slow-wave-dependent influx of CSF into perivascular spaces and the brain parenchyma, which, by causing an exchange of CSF and interstitial fluid, may be involved in clearing the brain of waste products. However, as the quantitative impact of these stronger brain pulsations in N3 versus N2 sleep is difficult to assess directly with our study design, it should be mentioned that the duration of N2 in a typical night's sleep is about twice as long as that of N3 sleep, potentially leaving more time for brain clearance. This leaves open the question whether the cumulative time spent in NREM sleep or the total slow wave amount is more important for glymphatic function.

Although smaller than during sleep, we also noted an increase in respiration-driven brain pulsations in GM and WM during sleep-deprived wakefulness. We speculate this may indicate increased CSF influx to the brain due to elevated LFPs during sleep deprivation, or alternatively, it may highlight the link between brain fluid dynamics and EEG slow waves, given the slight rise in awake EEG delta power after 35 h of wakefulness.

Preclinical studies suggest that central inhibition of noradrenergic receptors increases CSF influx [8,54] and brain clearance [55]. However, as our participants received only a single peroral dose of carvedilol, which is known to enter the brain in minimal amounts [47,56], we believe that its main pharmacological effects were on the cardiovascular system. Consistent with this, we observed reductions in blood pressure, plasma norepinephrine, and cardiac-driven brain pulsations, alongside the previously discussed suppression of LFPs. Accordingly, we suggest that the carvedilol-associated attenuation of cardiac-driven brain pulsations reflects its negative inotropic effect on cardiac contractability, which in turn dampens the heartbeat-induced pressure waves propagating through the vasculature and into the brain. Further supporting a primarily peripheral effect of carvedilol in our study, we found no discernible differences in sleep patterns between placebo and carvedilol sessions. The strength of respiration-driven brain pulsations observed under the placebo condition was largely replicated with carvedilol. Altogether, these findings mirror preclinical evidence that systemic administration of adrenergic agonists increases cortical artery pulsatility [20,24].

Our study is not without limitations. To eliminate the influence of menstrual cycle on sleep and circadian rhythm [57], we only included males participants. While preclinical studies have not found any male/female differences in glymphatic flow [58], this limits the generalizability of our findings. Additionally, the study was conducted under tightly controlled conditions and included only young, healthy individuals confirmed to be good sleepers, which further warrants caution when extrapolating the results to broader populations. Moreover, given the coarse spatial resolution of MREG, we cannot distinguish CSF in perivascular spaces from CSF and water in the brain parenchyma, and further chose to assess CSF space pulsations in the lateral ventricles to minimize partial volume effects from adjacent tissues. As a result, our data do not allow for examination of CSF pulsations in the fourth ventricle, as has been done by other studies cited here. Regarding LFPs, there is currently no consensus in the literature on their appropriate frequency range. We chose a range consistent with B-waves (0.01–0.034 Hz), whereas others have applied a broader frequency band (~0.01–0.1 Hz) [28,39]. In a posthoc analysis, we assessed whether using a 0.01–0.1 Hz band would alter our results and found them consistent with the band used in our study (see Supporting information; S1 File). Finally, while we attempted to appropriately adjust for multiple testing, we acknowledge that the number of analyses increases the risk of type I errors. Nonetheless, this is the first study to carefully control for effects of sleep pressure, time of day, circadian rhythm, and alcohol and caffeine intake when evaluating how sleep deprivation, vigilance state and sleep depth influence LFPs as well as respiration- and cardiac-driven brain pulsations.

## Conclusion

We show that heightened sleep pressure and slow-wave-rich sleep exert distinct effects on cerebral vasomotion and on the strength of brain pulsations induced by the respiratory and cardiac cycles.

Our investigation unveils that following 35 h of wakefulness, MR-detected LFPs intensify throughout the entire brain, and that objective measures of sleep pressure exhibit a positive correlation with the magnitude of LFPs during sleep. Additionally, our study suggests that pharmacological vasodilation of the cerebral vasculature affects LFPs, providing support for the notion that LFPs originate from vasomotion. Because cerebral vasomotion is thought to be a key driver of CSF flow in the brain, these important findings offer crucial insights into how the vasomotor system responds to sleep deprivation, potentially serving as a means to mitigate the accumulation of brain waste products. Furthermore, we illustrate that during NREM sleep stages N2 and N3, respiration- and cardiac-driven brain pulsations intensify in both GM and WM, with strength increasing as sleep deepens. As the strength of these brain pulsations likely reflect brain fluid motion and brain water and CSF content, our observations align with preclinical evidence of a slow-wave-dependent influx of CSF into perivascular spaces and the brain parenchyma. These new findings collectively contribute to our understanding of the intricate dynamics of homeostatic sleep mechanisms, vasomotor activity, brain fluid flow and the clearance of waste products from the brain.

## Materials and methods

### Ethics statement

Approval was granted by the Danish ethics committee of the Capital Region of Denmark (journal ID: H-16045933), and the study adhered to the principles of Declaration of Helsinki. Written informed consent was obtained from all participants before enrollment in the studies.

### Study design

We conducted a circadian-controlled sleep and sleep deprivation study that included a double-blind, placebo-controlled, cross-over administration of the nonselective β1-, β2- and α1- adrenergic antagonist carvedilol (Fig 1). The study was preregistered at ClinicalTrials.gov (NCT03576664) and conducted at Rigshospitalet in Copenhagen (Denmark), between January 2018 and May 2019.

During an adaptation week and throughout the study period, participants followed a strict 8 h sleep, 16 h wake protocol. Sleep–wake rhythms were monitored continuously with actigraphy (Actiwatch spectrum, Philips Respironics) and a sleep diary to ensure that all measurements were performed during the same circadian phase and at comparable sleep propensities. Furthermore, participants were not allowed to drink alcohol, and daily caffeine consumption was limited to two or less cups of coffee (≤200 mg caffeine/day) taken no later than 2 pm. Three EEG/MRI scan sessions were conducted on separate days in the evening (standardized time: ~18.30–20.30): A well-rested scan scheduled after 11 h of wakefulness followed by two sleep-deprived scans performed after 35 h of wakefulness. Sleep-deprived scans were performed 1 week apart. Participants received either placebo or 25 mg of carvedilol 1 h before initiation of sleep-deprived scans, as carvedilol reaches peak plasma concentration within 1–2 h [46]. Placebo and carvedilol were administered in identical capsules (manufactured and distributed by Capital Region Pharmacy) and participants were randomized to either the placebo-carvedilol or carvedilol-placebo sequence (10 in each) by a research administrator not otherwise involved in data collection or analysis. Participants and all members of the research team remained blinded throughout the entire study. All three EEG/MRI scan sessions included a period of wakefulness (well-rested scan: ~45 min; sleep-deprived scans: ~30 min) followed by an 1-h sleep opportunity. During the awake period, lights were on, and participants were continuously monitored to ensure they remained awake with eyes open. During sleep opportunities, lights were turned off, the scanner

environment was kept as quiet as possible, and participants were told to relax and that they were allowed to sleep. Participants' blood pressure was measured before and immediately after scan sessions and a blood sample to quantify plasma norepinephrine level was collected ~15 min after scans had ended. Sleep pressure was assessed with PVT performance before and after all scan sessions.

During the nights leading up to each of the three study periods and during the two nights following the sleep deprivation scans, participants slept in a controlled sleep environment monitored with polysomnography (PSG). The prolonged wakefulness periods were monitored with a minimal EEG setup and participants were continuously supervised by members of the research team. The only deviation from the original experimental protocol (S2 File) is that, due to issues with scanner availability, the schedule was shifted 3 h later than originally planned.

### Study population

Twenty healthy males, aged 18–29 years, were included in and completed the sleep study (see consort flow diagram in Fig 2). Participants were recruited from a local database of individuals interested in participating in brain imaging studies and via a national test subject recruitment database (www.forsøgsperson.dk). All participants were right-handed, fluent in Danish, had normal blood pressure, reported no sleep issues when questioned systemically, and had no prior or current neurological or psychiatric disorders, learning disabilities, severe somatic diseases, or any use of prescription drugs of relevance for the study. All were nonsmokers and did not have excessive use of alcohol or illicit drugs and had not performed shiftwork three months prior to study participation. Participants who had crossed two or more time zones were required to have at least 14 days of recovery per time zone before enrollment. Before inclusion, participants underwent a PSG screening night to rule out undiagnosed sleep disorders or low sleep efficiency (defined as <85%).

### Polysomnography and prolonged wakefulness EEG recordings

All sleep and wakefulness EEGs recorded outside the MR environment were recorded using the battery powered and transportable SomnoScreen Plus system (Somnomedics, Germany). Impedance values were kept below 6 kΩ at the beginning of all recordings.

Overnight PSG recordings were performed in private rooms with blinded windows. Data across 18 EEG electrodes, placed according to the internationally standardized 10–20 system [59], were recorded simultaneously with submental electromyogram, electrooculogram (EOG), and electrocardiograph (ECG). Overnight sleep periods were defined by participants blinking 10 times at lights-off and again at lights-on. Participants were prescribed fixed lights-off/on times corresponding to 8-h (standard night) and 10-h (recovery night) sleep opportunities, and data analysis was restricted to a maximum of 8 h (480 min) to enable comparisons between nights.

Vigilance during the two 35-h periods of prolonged wakefulness was monitored using an EEG setup comprising six EEG electrodes and EOG. EEG signals were recorded continuously throughout both wakefulness periods, with the recorders placed in a bag to allow participants to move freely.

A single expert, blinded to treatment condition, visually scored the EEG recordings in 30-s epochs according to standard AASM criteria (American Academy of Sleep Medicine, 2007), taking artifact rejection and filter settings into consideration. Scoring was performed using DOMINO software (Somnomedics, Germany).

### MRI acquisition

Simultaneous EEG and MRI was performed on a Siemens (Erlangen, DE) MAGNETOM 3T Prisma scanner with a 64-channel head coil. Structural images were acquired using a high-resolution, whole-brain, T1-weighted MPRAGE scan with the following parameters: Inversion time = 900 ms, repetition time = 1,900 ms, echo time = 2.52 ms, flip angle = 9°, in-plane matrix = 256 × 256, in-plane resolution = 0.9 × 0.9 mm, slices = 208, slice thickness = 1.0 mm. For accelerated

functional MRI imaging, we used a MREG sequence obtained from the University of Freiburg [60]. MREG allows for 3D whole-brain imaging with a repetition time of 100 ms at a resolution of 3 × 3 × 3 mm using a stack-of-spirals k-space under-sampling trajectory. Further scan parameters were: 3D matrix size = 64 × 64 × 50, echo time = 33 ms, flip angle = 25°, field of view = 150 mm, and a gradient spoiling of 0.1 (for $n$ = 3, gradient spoiling was 1.0) [37].

Participants were fitted with an MRI-compatible EEG-cap as well as a respiration belt and a pulse oximeter wired directly to the scanner. They also wore earplugs to reduce scanner noise, and head cushions were used to restrict head movement. During sleep opportunities, scans alternated between 5-min MREG and 5-min multiband echo-planar sequences (the latter data not reported here) to maintain a steady noise level.

### MREG and structural brain data preprocessing

MREG data were reconstructed with a MATLAB reconstruction tool provided by the sequence developers [60], using L2-Tikhonov regularization with lambda = 0.2 and a regularization parameter determined by the L-curve method. To ensure steady-state signal saturation, the first 10 s of data (= 100 images) were excluded from all MREG scans. Next, MREG data were preprocessed and analyzed using Statistical Parametric Mapping software (SPM12, Welcome Trust Center for Neuroimaging, UCL) in MATLAB (R2017b, The Mathworks), including motion correction, which was performed by realignment and co-registration of the high-resolution T1 structural image to the MREG data. Co-registration was enhanced with FSL's Brain Extraction Tool, employing a fractional intensity threshold of 0.3 on the first functional volume of each MREG scan, to which the structural image was co-registered. Subsequently, the co-registered structural images were segmented into GM, WM, and CSF maps.

In order to evaluate CSF pulsatility with as little interference of surrounding tissue and vascular pulsations as possibly, we created a CSF ROI in the left lateral ventricle: We marked a sphere with a radius of 2 cm around a manually selected midpoint of the left ventricle defined in MNI space (MNI coordinates: (8,−10,24)), keeping only the voxels with a segmentation-based CSF probability larger than 0.5. The ventricle ROI were then normalized to each scan session and used to asses CSF pulsations in tissue-type analyses.

### EEG acquisitions during MRI

To monitor sleep and wakefulness during scans, participants wore a MRI-compatible EEG cap (Electrical Geodesics, Eugene, OR) with 256 EEG channels and a single reference electrode (Cz). The cap was selected based on participants' head circumference and prepared with an electrolyte/shampoo solution. Cap placement was aided by positioning the reference electrode at the vertex and visual inspection of predetermined reference points. The cap was secured with a net and kept moist by a shower cap. Impedances were kept below 50 kΩ. Two ECG electrodes were placed on the left side of the sternum, at the 4th and 5th intercostal spaces. Optical cables from the cap were routed along the side of the head inside the MRI head coil. EEG and ECG data were acquired with a 1 kHz sampling rate and all cables were connected to a synchronization box to match acquisition to the MR scanner clock frequency.

### Removal of MR-induced artifacts on EEG

Removal of gradient and ballistocardiographic artifacts from the EEG was performed with MATLAB (R2014a, The MathWorks) using standard methods in a stepwise approach. First, the Average Artifact Subtraction method was used to eliminate gradient artifacts. This creates a template gradient artifact by using the MR-trigger signal as a time-locking event and then averaging across the nearest 30 artifacts, in a moving window on high-pass filtered data [61].

Second, the Optimal Basis Sets approach was used to adaptively remove ballisto-cardiographic artifacts over time. This approach uses the ECG R-peak as a time-locking event and combines the local moving average template construction with a combination of basis functions, derived from a principal component analysis of the gradient cleaned EEG-signal [62].

## Sleep scoring of EEG during MRI

EEG recordings were visually scored in 30-s epochs in accordance with standard AASM criteria (American Academy of Sleep Medicine, 2007) using DOMINO scoring software (Somnomedics, Germany). To accurately quantify sleep in the MR environment, EEG recordings were scored by two independent experts blinded to subject, treatment, and scan type (lights on or lights off). In addition, scorers were also instructed to mark epochs with residual MR artifacts and aberrant nonphysiological data as "artifacts". The overall agreement between the two scorers was 73%, which is similar to what would be expected for standard sleep scorings in a nonMR environment [63].

Final sleep staging for EEG epochs was determined by consensus between experts as follows: 30-s epochs were labeled as "wakefulness" if they occurred before the first onset of NREM sleep stages N2 or N3 (scored by either expert) and if both experts agreed on "wakefulness" or if one scored it as "wakefulness" and the other scored it as "artifact." Epochs were classified as "NREM sleep" if both experts scored them as either NREM sleep stage N2 or N3, while epochs were only classified as "N2 sleep" or "N3 sleep" if both experts scored them as such.

N1 sleep was not included in analysis due to a limited number of epochs scored as such ($n = 32$, with 11 originating from a single participant).

Delta power ratio was quantified for all 30-s epochs, irrespective of scored vigilance state, as the ratio between the computed EEG band power ($mV^2$) in the 1–4 Hz range and in the 0.5–30 Hz range.

## Quantitative analysis of EEG during MRI

For quantitative analysis of EEG, data were cleaned subject- and condition-wise by rejecting channels exceeding a kurtosis threshold of 5 using EEGlab's pop_rejchan function [64]. The leads C3-M2 and C4-M1 were constructed either from preselected channels ($Chan_{\{C3\}} = 59$, $Chan_{\{C4\}} = 183$, $Chan_{\{M1\}} = 94$, $Chan_{\{M2\}} = 190$) or, if these were rejected, an average of the neighboring channels. Artifact segments were rejected from the two leads in 0.5 s windows with 0.25 s overlap if the power in the 35–120 Hz range exceeded 15 dB.

Delta power ratio was quantified for all 30-s epochs, irrespective of scored vigilance state, in both leads using MATLAB's bandpower function on epoch data from which artifact segments had been rejected. Bandpower ($mV^2$) was computed for the 1–4 and 0.5–30 Hz ranges, with relative delta-power defined as the ratio of the two. The reported delta power ratio is a mean of the two leads. Thirty-second epochs classified as "artifact" by either of the two EEG-scorers were excluded from analysis of EEG delta power ratio.

## MREG data analysis

For analysis of LFPs, we used the full 5-min MREG sequences and assessed spectral power within a frequency range based on the clinically described B-waves (0.5–2 waves pr. min) [65]. For each 5-min MREG scan (corresponding to 10 scored EEG epochs), data were high-pass filtered with a 10-order IIR filter (highpass 0.008 Hz). Spectral analysis was then performed for individual voxels in a whole-brain mask consisting of all voxels with either a GM, WM, or CSF probability larger than 0.1, using the MATLAB periodogram function with a Hanning window and bin width of 0.00244 Hz. Next, the voxel-wise MREG spectra were averaged to create respective whole-brain or tissue-type-specific MREG power spectra. Five-min scans with framewise displacement >3 mm in one or more 30-s epoch were excluded from analysis. To avoid spill-over effects from the bandpass filtering, the exact frequency interval for LFP analysis was set to 0.01221–0.03418 Hz (~0.7–2.1 waves pr. min). Five-min MREG scans were selected for LFP-analysis if EEG experts had classified at least 80% (8/10 epochs) of the simultaneously recorded EEG epochs as either wakefulness or NREM sleep and only if they occurred in lights-on or lights-off conditions, respectively.

For analysis of respiration- and cardiac-driven brain pulsations, we used a 30-s dataset and assessed spectral power in individually tailored respiration and cardiac frequency bands. Data from all 5-min MREG scans were high-pass

filtered with a 10-order IIR filter (passband 0.1 Hz), and segmented into 30-s epochs, which were temporally aligned with corresponding sleep-scored EEG epochs. For each 30-s epoch, spectral analysis was performed as described for the 5-min dataset, but with a bin width of 0.02 Hz. Epochs with a framewise displacement above 3 mm were excluded ($n = 91$ epochs). We defined epoch-by-epoch respiration and heart rate frequencies using temporally aligned data from the respiration belt and the pulse oximeter, imported using the TAPAS PhysIO toolbox [66]. The dominant peaks in respiration (0.13–0.5 Hz; 7.8–30 min$^{-1}$) and cardiac frequency ranges (0.67–2 Hz; 40.2–120 min$^{-1}$) were determined from the periodogram-determined power spectrum of each 30-s epoch of physiological data. The epoch-by-epoch peak frequencies for respiration and heart rates were then used to define the individually tailored 30-s epoch-specific respiration and cardiac frequency bands in the MREG spectra. Frequency ranges for the epoch-wise respiration and cardiac bands were defined as the epoch-wise respiration rate ± 0.06 Hz (= ±3 bins), and the epoch-wise heart rate ± 0.1 Hz (= ±5 bins), respectively. Thirty-second epochs with nonphysiological values (<40 or >90 heartbeats per min. and/or <8 or >25 breaths per min.) were excluded. Thirty-second MREG epochs were selected for sleep- and sleep deprivation analyses if experts had classified the corresponding EEG as wakefulness, NREM sleep, N2 sleep, or N3 sleep. For EEG delta power analysis, MREG epochs were only included if neither of the experts had scored the corresponding EEG as "artifact".

## Psychomotor vigilance

A PVT (e-Prime software, Psychology Software Tools , Pittsburgh) was used to assess cognitive effects of prolonged wakefulness [67]. PVT is a simple reaction time task in which participants press the space key as quickly as possible upon seeing a digital millisecond counter on the computer screen. A total of 100 stimuli are presented at random inter-stimulus intervals of 2–10 s. Participants completed a training session during the adaptation week and performed the PVT before (rested: 7-h awake; sleep-deprived: 31-h awake) and immediately after scans (following an 1-h sleep opportunity) (Fig 1). Two validated PVT variables were quantified [48,49]: "median reaction time" and "lapses of attention (reaction time > 500 ms). Impaired psychomotor vigilance due to sleep deprivation was evaluated by (1) quantifying the increase in median reaction time from rested wakefulness (measure before sleep-deprived scan—measure before well-rested scan) and (2) the absolute number of lapses of attention before the sleep-deprived scan (Freeman-Tukey transformed: $\sqrt{x} + \sqrt{(x + 1)}$). Improvements in psychomotor vigilance due to sleep in the scanner were computed as the difference between measures taken before and immediately after scans.

## Statistical analyses

MREG spectral power within the LFP, respiration, and cardiac frequency bands (calculated as the sum of spectral power within the respective frequency ranges) were considered the primary outcome measures, reflecting low frequency cerebral vasomotion, respiration-driven brain pulsations and cardiac-driven brain pulsations, respectively.

MREG spectral power in the three frequency bands from the well-rested (awake scan), sleep-deprived placebo (awake and sleep scan), and sleep-deprived carvedilol scan sessions (awake and sleep scan) was log transformed to mitigate skewness in the distribution before being analyzed using linear mixed models. Linear mixed models included sleep deprivation (well-rested versus sleep-deprived), vigilance state (awake versus NREM sleep), sleep depth (sleep-deprived awake versus N2 versus N3), tissue type (gray matter versus WM versus CSF), treatment (placebo versus carvedilol; randomized to sleep deprived scan session 1 and 2), PVT measurements ("median reaction time" and "lapses of attention") and delta power ratio as additive fixed effects, with interactions between vigilance state and tissue type as well as between vigilance state and treatment. Subject ID, period effect (well-rested session, sleep-deprived session 1, sleep-deprived session 2), and scan type (lights on, lights off) were included as random intercepts with sleep opportunity being nested into scan session itself nested into subject ID. To account for vigilance-state-related changes in peripheral cardiorespiratory signals as well as the impact of declining spectral power as a function of higher frequencies [68], respiration or heart rates recorded simultaneously with MREG-scans were included as fixed effects in all models

assessing spectral power within respiration or cardiac frequency bands, respectively. *P*-values for testing fixed effects were obtained using Wald tests. The expected log spectral power at various conditions (e.g., sleep-deprived under carvedilol) was computed from the mixed model estimates to illustrate the model fit. Lastly, in a sensitivity analysis, we included the period effect as a fixed effect rather than a random intercept in the linear mixed model described above.

Associations between (*i*) PVT measurements and mean log spectral power within LFP, respiration, and cardiac frequency bands in 5-min scans/30-s epochs and (*ii*) delta power ratio and log spectral power within respiration and cardiac frequency bands in 30-s epochs were assessed with Wald tests obtained from linear mixed models (denoted $p_{adj}$). Corresponding correlations coefficients were deduced from mixed model estimates (denoted $r_{adj}$). For reference, Pearson's correlation coefficients for associations between (*i*) and (*ii*) were also evaluated (regardless of subject ID, scan session, sleep opportunity, respiration and heart rates, and treatment) and denoted $r_{raw}$.

Where pairwise testing was appropriate, we conducted Student's two-tailed *t* tests for data with a normal distribution and Wilcoxon signed-rank tests for data not normally distributed. *P*-values equal to or below 0.05 were considered significant except in analyses where three levels of a fixed effect were evaluated (and compared pairwise). Here, *P*-values were adjusted for multiple comparison using Bonferroni correction. If not otherwise specified, estimates are reported as mean SEM. All analyses were performed with the statistical software R (http://www.R-project.org/). The R packages lme4 (version: 1.1.29), lmerTest (version: 3.1.3), and LMMstar (version: 0.8.9) were used for, respectively, fitting random intercept models, performing Wald tests, and estimating expected means and correlation parameters.

For detailed descriptions of all statistical models applied throughout the paper and power calculations, see *Supporting information* (S1 File).

## Supporting information

**S1 Fig. EEG-recorded vigilance states during MR-EEG sessions.** Boxplots illustrate **(A)** absolute and **(B)** relative time spent in wakefulness and NREM sleep stages N1, N2, and N3 during rested and sleep-deprived (placebo and carvedilol) scan sessions. All 30-sec EEG epochs recorded simultaneously with MR scans were included in analyses. EEG-epochs were included in analyses when the two independent EEG scorers agreed on staging. Epochs with scorer disagreement and epochs scored as artifacts were categorized as "A". There is no difference between carvedilol and placebo conditions (Student's paired *t* test, $p_{all} > 0.05$). Box-plot elements include: median (center line), upper and lower quartiles (box limits), 1.5× interquartile range (whiskers), and outliers (points). $N = 20$.
(TIFF)

**S2 Fig. Plasma norepinephrine (pg/mL) immediately after all scans.** Normal range: 200–1,700 pg/mL. *p*-values are from a linear mixed model have been adjusted for multiple comparisons with Bonferroni correction Box-plot elements include: median (center line), upper and lower quartiles (box limits), and 1.5× interquartile range (whiskers). $N = 20$.
(TIFF)

**S3 Fig. Participant-wise correlations between EEG delta power ratio as recorded simultaneously with MREG-imaging, and spectral power.** Each dot represents a 30-s epoch and each participant's data is shown with a unique color. $r_{est}$ and $p_{est}$ are estimated correlation coefficient and estimated *p*-value from linear mixed models, where repeated measurements are taken into account. **(A)** A positive slope was observed for 17 of 19 participants in the respiration frequency band, and of these 13 were statistically significantly positively correlated. **(B)** For the cardiac band, 16 of 20 participants were positive and 11/20 had a significant positive slope.
(TIFF)

**S4 Fig. Effects of sleep deprivation and NREM sleep (combined N2 and N3) on brain pulsations across gray matter, white matter, and CSF.** Results from sensitivity analyses of the effects of sleep deprivation and NREM sleep

on MREG power in the **(A)** LFP, **(B)** respiration, and **(C)** cardiac frequency bands across three tissue types (gray matter, white matter and CSF). Error plots represent estimates (estimated means ± SEM) from linear mixed models, run separately for each tissue type.
(TIFF)

**S1 Table. Wakefulness period leading up to MR/EEG scan sessions.**
(DOCX)

**S2 Table. Standardized sleep during nights before study days.**
(DOCX)

**S3 Table. 30-s dataset for evaluation of respiration- and cardiac-driven brain pulsations.**
(DOCX)

**S4 Table. 5-min dataset for evaluation of LFPs.**
(DOCX)

**S5 Table. 30-s dataset for evaluation of sleep depth effects.**
(DOCX)

**S6 Table. Recovery nights after sleep deprivation.**
(DOCX)

**S1 File. Supplementary materials.**
(DOCX)

**S2 File. Experimental protocol.**
(PDF)

## Acknowledgments

We would like to thank Helle Leonthin for sleep staging and Emily Beaman, Cecilie Lerche Nordberg and members of the Center for Translational Neuromedicine lab for their assistance with data collection and supervision of proper sleep deprivation. We also thank Dan Xue from the Center for Translational Neuromedicine for assistance with illustrations.

## Author contributions

**Conceptualization:** Sebastian Camillo Holst, Poul Jørgen Jennum, Maiken Nedergaard, Gitte Moos Knudsen.

**Data curation:** Sara Marie Ulv Larsen, Sebastian Camillo Holst, Anders Stevnhoved Olsen, Dorte Bonde Zilstorff, Pia Weikop, Simone Pleinert.

**Formal analysis:** Sara Marie Ulv Larsen, Anders Stevnhoved Olsen, Dorte Bonde Zilstorff, Kristoffer Brendstrup-Brix.

**Funding acquisition:** Sebastian Camillo Holst, Vesa Kiviniemi, Maiken Nedergaard, Gitte Moos Knudsen.

**Investigation:** Sara Marie Ulv Larsen, Sebastian Camillo Holst, Dorte Bonde Zilstorff, Simone Pleinert.

**Methodology:** Sara Marie Ulv Larsen, Sebastian Camillo Holst, Anders Stevnhoved Olsen, Brice Ozenne, Pia Weikop.

**Project administration:** Sara Marie Ulv Larsen, Sebastian Camillo Holst, Dorte Bonde Zilstorff.

**Resources:** Sebastian Camillo Holst, Gitte Moos Knudsen.

**Software:** Anders Stevnhoved Olsen, Brice Ozenne, Vesa Kiviniemi, Poul Jørgen Jennum.

**Supervision:** Poul Jørgen Jennum, Maiken Nedergaard, Gitte Moos Knudsen.

**Visualization:** Sara Marie Ulv Larsen, Anders Stevnhoved Olsen, Brice Ozenne.

**Writing – original draft:** Sara Marie Ulv Larsen, Sebastian Camillo Holst, Gitte Moos Knudsen.

**Writing – review & editing:** Anders Stevnhoved Olsen, Brice Ozenne, Dorte Bonde Zilstorff, Kristoffer Brendstrup-Brix, Simone Pleinert, Vesa Kiviniemi, Poul Jørgen Jennum, Maiken Nedergaard.

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
