## [Editor Report · Decision Letter 0]

15 Jan 2025

Dear Dr Knudsen,

Thank you for submitting your manuscript entitled "Sleep deprivation promotes cerebrovascular oscillations while respiration- and cardiac-driven brain pulsations escalate with sleep intensity" for consideration as a Research Article by PLOS Biology.

Your manuscript has now been evaluated by the PLOS Biology editorial staff as well as by an academic editor with relevant expertise and I am writing to let you know that we would like to send your submission out for external peer review.

Once your full submission is complete, your paper will undergo a series of checks in preparation for peer review. After your manuscript has passed the checks it will be sent out for review. To provide the metadata for your submission, please Login to Editorial Manager (https://www.editorialmanager.com/pbiology) within two working days, i.e. by Jan 17 2025 11:59PM.

Kind regards,

Luke

Lucas Smith, Ph.D.

Senior Editor

PLOS Biology

lsmith@plos.org

---

## [Decision Letter · Decision Letter 1]

17 Mar 2025

Dear Dr Knudsen,

Thank you for your patience while your manuscript "Sleep deprivation promotes cerebrovascular oscillations while respiration- and cardiac-driven brain pulsations escalate with sleep intensity" was peer-reviewed at PLOS Biology. Apologies for the delay in sending our decision. It took a bit longer than usual to recruit suitable reviewers. Your manuscript has now been evaluated by the PLOS Biology editors, an Academic Editor with relevant expertise, and by several independent reviewers.

In light of the reviews, which you will find at the end of this email, we would like to invite you to revise the work to thoroughly address the reviewers' reports.

As you will see below, the reviewers find the study potentially interesting, but all reviewers have suggestions to strengthen the study further. Some of these points will require additional analyses.

Given the extent of revision needed, we cannot make a decision about publication until we have seen the revised manuscript and your response to the reviewers' comments. Your revised manuscript is likely to be sent for further evaluation by all or a subset of the reviewers.

**IMPORTANT - SUBMITTING YOUR REVISION**

*Re-submission Checklist*

*Published Peer Review*

*PLOS Data Policy*

*Blot and Gel Data Policy*

Sincerely,

Christian

Christian Schnell, Ph.D.

Senior Editor

PLOS Biology

cschnell@plos.org

on behalf of

Lucas Smith, Ph.D.

Senior Editor

PLOS Biology

lsmith@plos.org

REVIEWS:

Reviewer #1: see attached

Reviewer #2: This study investigated how the brain oscillations of different frequency ranges, measured by fast fMRI, after sleep deprivation. It focuses on brain pulsations of the three frequency bands: the low-frequency band (0.012 - 0.34 Hz), respiration and cardiac pulsations bands. It was found that the low-frequency oscillations (LFOs) increased during awake periods after sleep deprivation more than during NREM sleep, and LFOs during sleep are correlated with cognitive measures of sleep pressures via psychomotor vigilance test (PVT). In contrast, the brain pulsations of respiratory and cardiac ranges increased during NREM sleep and were correlated with EEG delta power. In addition, the study also showed that carvedilol, an adrenergic antagonist, dampens LFPs. Overall, this is an important and timely study with very interesting findings. Especially, it differentially linked LFO and respiration/cardiac pulsations to sleep pressure and NREM slow wave sleep. My specific questions and comments are listed below.

1. Why were the PVT measures linked to LFOs during sleep (Figure 3) but not during sleep-deprived awake sessions? Since the LFOs were linked to sleep pressure and had the peak amplitude during sleep-deprived awake whereas non-significant increase during sleep (Figure 2), it would make more sense to me to study the LFOs during sleep-deprived awake.

2. It would be great to provide some comparison for the sleep scoring between the rested and sleep-deprived awake conditions. According to the Methods section, 5-min scans with more than 80% time scored as "wake" were identified and used for LFO quantifications. The 80% looks like a high percentage that would guarantee the dominance of wake periods analyzed. However, if the light sleep stages (N1 and N2) are associated with much larger LFOs than wake, which could be the case as shown in Fultz et al., Science 2019, their biased presence in the 20% of analyzed data could significantly affects the results, particularly the LFO differences seen between the rested and sleep-deprived awake conditions.

Most importantly, I don't actually understand the necessity of using the 5-min scans. According to Table S4, only 12 out of 20 subjects have those scans in the sleep-deprived awake condition. The quantification of spectral power of LFOs shouldn't require the continuation of data. To me, it is completely OK to filter the data in the low-frequency range, cut them into 30-sec segments, concatenate those of the same wake/sleep conditions, and then quantify their spectral power, even the 30 seconds are shorter than the LF range (0.012 - 0.034 Hz).

3. Why aren't there any results for N1 sleep stage, which could be critical?

4. Should the upper frequency limit for LFOs be 0.034, as shown in Figure 2, rather than 0.34, which appear in several places in the paper?

5. Are 34 data points (14 for the placebo group whereas 17 carvedilol group) in Figure 3 due to the availability of 5-min N2/N3 sleep data? Again, my comment above applies to here. The power of LFOs can be assessed using 30-sec epochs.

6. The associations between the brain pulsations of respiratory and cardiac frequencies and delta power ratio (Figure 4F-G) are largely overlapped with their differences seen between awake and sleep (Figure 4A-D). The associations are largely driven by the differences between data points from awake (blue) and sleep (orange) conditions, which are known to have significantly different delta power. If the purpose of this association analysis was to further emphasize the link between these pulsations with slow wave activity, it should be conducted within sleep condition. In addition, what is the rationale of excluding the sleep-deprived awake from this analysis?

7. Negative results were mentioned but not shown in a few places, e.g., lines 408-410 in page 15, lines 356-357 in page 13, lines 217-219 in page 8. It would be more appropriate to provide those data as supplementary materials if they are less important.

8. The effect of carvedilol on LFOs are interpreted as compelling evidence suggesting that LFOs "originate" from cerebrovascular oscillations. Is this to imply the non-neural physiological nature of LFOs? Can we really say that? Like traditional fMRI techniques, the MREG measures BOLD effect that is essentially vascular. Even LFOs originate from neural activity, carvedilol is expected to have effects on them by affecting coupled vascular responses. Moreover, there are increasing evidence suggesting that the cerebral vasomotion are associated with low-frequency neural activity. For fMRI LFOs, recent studies also suggested they may take highly organized pattern as waves propagating across cortical hierarchies, and the existence of whole-brain low-frequency brain activity at single-neuron level. Therefore, it is important to clarify the interpretation and implication of the findings regarding carvedilol's effect.

9. Should there be "control" frequency bands to show that changes in brain pulsations in the respiratory and cardiac ranges are indeed related to these two physiological processes? One reason I ask about this is that the cardiac frequency range is overlapped with that for slow waves (0.5-4 Hz), which are significantly implicated as discussed in the Discussion section. Thus, it could be important to find out if similar changes are seen in the frequency ranges that are either between the respiratory and cardiac rhythms, i.e., 0.4-0.5 Hz, or higher than typical heart rate at rest, e.g., 2-4 Hz?

Reviewer #3: Ulv Larsen and colleagues present the results of a sleep study with a nested randomised crossover trial. I was asked to review the paper due to the inclusion of the randomised trial; my particular area is statistics in clinical trials.

Firstly, I would say that this seems like a well-conducted study (if a little hard to follow at times, for a non-specialist), and the analyses appear to have been done quite well. I will have some comments on the statistics (I cannot help myself), but these are mainly to do with aspects of the presentation.

However, I will start with the randomised crossover trial element of the study. This is a 2x2 crossover trail of carvedilol vs. placebo. The facts that it was double blind, that the randomisation was done by someone not otherwise involved in the trial, and that there was a washout period between treatment phases, makes this a well-designed and conducted trial.

The first place I usually start with any trial is the sample size justification. It took me a while to find it, hidden away at the end of one of the supplements, and I would have put it in the main methods section of the paper. The calculation is reproducible, which is a good sign. However, the primary outcomes of the trial are "MREG spectral power within the LFO, respiration and cardiac frequency bands" (line 603), compared at two time points (when awake, and during NREM sleep; Figure 6), suggesting a total of 6 analyses considered "primary" for the trial. Therefore, whist the study had 85% power to detect differences at p<0.01, there was only 83.9% power at a more robust p<0.00833 (i.e. 0.05/3). Still, that is still quite good power, so the sample size is fine. My point is that none of the between-treatment differences shown in Figure 6 meets either of these significance thresholds. So, it could be argued that these results, whilst suggestive of treatment effects, do not provide "compelling evidence supporting the hypothesis that LFOs originate from cerebrovascular oscillations" (lines 330-331). Perhaps the authors should be a little less enthusiastic about the crossover trial results.

Otherwise, the crossover trial is quite well analysed, though with a rather elaborate mixed effects model. Personally, I would adopt a simpler model and look at the sensitivity of the results to alternative methods or further adjustments, but what has been done is fine. That said, I would be explicit about including a fixed effect for period effects; this may be already be done as a side effect of including random effects for scan sessions, but it is not immediately obvious.

Beyond the randomised trial, a lot of analyses are presented looking at changes in different parameters between different time points during the study, and correlations between different measures. There is a tendency to adopt language that implies that the associations observed are causal. Whilst many of these assertions may well be true, given the quite tightly controlled conditions in the study, it does not automatically apply that an association observed in a non-randomised, unblinded study, is a causal association. Any statement of cause and effect is, to some degree, taking a leap from the data to the interpretation, so a little caution is recommended.

The sheer number of analyses done in this study does lay it open to type 1 error, and this could perhaps be recognised as a limitation.

Finally, and I accept this is a particular bugbear of mine, I have never liked the use of the standard error as a descriptive statistic. When the aim is to illustrate a distribution (of a roughly Normal measure), then mean and standard deviation should be used. When the aim is to illustrate the precision of an estimate, then the estimate (e.g. mean, or mean difference) and a 95% confidence interval is a better option. My own personal feeling is that the use of estimate +/- SE, either in text, or particularly in figures, is a way of making differences appear more striking. However, I know this is common practice away from my own sphere of statistical analysis, so I am not going to insist that the authors change what they have done.

---

## [Decision Letter · Decision Letter 2]

11 Aug 2025

Dear Dr Knudsen,

Thank you for your patience while we considered your revised manuscript "Sleep deprivation promotes cerebrovascular oscillations while respiration- and cardiac-driven brain pulsations escalate with sleep intensity" for consideration as a Research Article at PLOS Biology. Your revised study has now been evaluated by the PLOS Biology editors, the Academic Editor and the original reviewers.

The reviews are appended below (and reviewer 1's comments are attached). You will see that both reviewers 2 and 3 are fully satisfied by the changes made in this revision, and reviewer 1 agrees that the study has been strengthened. However, reviewer 1 has a number of important lingering concerns which will need to be thoroughly addressed before we can consider your study for publication. We would like to emphasize the need to refine the claims and terminology in the paper to improve clarity and to avoid overstating findings, as reviewer 1 suggests.

In light of the reviews, we are pleased to offer you the opportunity to address the remaining points from the reviewer 1 in a revision that we anticipate should not take you very long. However, we understand that you may need to provide additional analyses to address some of reviewer 1's comments - and so if you need more time, please do let us know.

We will assess your revised manuscript and your response to the reviewers' comments with our Academic Editor aiming to avoid further rounds of peer-review, although we might need to consult with the reviewers, depending on the nature of the revisions.

**IMPORTANT - SUBMITTING YOUR REVISION**

*Resubmission Checklist*

*Published Peer Review*

*PLOS Data Policy*

*Blot and Gel Data Policy*

Sincerely,

Luke

Lucas Smith, Ph.D.

Senior Editor

PLOS Biology

lsmith@plos.org

REVIEWS:

Reviewer #1: See attached

Reviewer #2: The authors have adequately addressed my comments, and I have no further questions.

Reviewer #3: The authors have addressed my concerns, so I would be happy to recommend accepting the paper

---

## [Editor Report · Decision Letter 3]

8 Oct 2025

Dear Dr Knudsen,

Thank you for your patience while we considered your revised manuscript "Sleep deprivation promotes cerebral vasomotion while respiration- and cardiac-driven brain pulsations escalate with sleep intensity" for publication as a Research Article at PLOS Biology. This revised version of your manuscript has been evaluated by the PLOS Biology editors and the Academic Editor.

The Academic Editor is fully satisfied by the changes made in response to the last round of review, and based on our his/her assessment of your revision, we are likely to accept this manuscript for publication. However before we can accept your study we need to you address a number of data and other policy-related requests in a last revision which we anticipate will not take very long. These are detailed below.

**IMPORTANT: Please address the following editorial requests.

1) TITLE: We have spent some time trying to come up with a title that will be more accessible to a broad audience, while still capturing the key messages of your paper, and we landed on the following suggestion:

"Sleep deprivation and sleep intensity exert distinct effects on brain pulsations driven by vasomotion or by respiratory and cardiac activity"

^If you agree (and feel this is supported) we suggest you adopt this title - but we are open to discussing this further.

2) ETHICS STATEMENT: Please update your ethics statement to include information about the form of consent (written/oral) given for research involving human participants. (I see this information is included later in the methods section, but it would be good to also include it in the ethics statement).

3) FINANCIAL DISCLOSURES: Please update your financial disclosures statement, in our editorial manager system, to include a link to the funders' websites.

4) DATA: I see your data availability statement currently says "Upon publication, all data underlying the results of this paper will be made available in an anonymized format through supporting information files." We ask that you please do go ahead and make your data available at this stage, and update your data availability statement and figure legends accordingly.

For details on the PLOS Data Policy, which requires that all data be made available without restriction, see here: http://journals.plos.org/plosbiology/s/data-availability. For more information, please also see this editorial: http://dx.doi.org/10.1371/journal.pbio.1001797

a. Supplementary files (e.g., excel). Please ensure that all data files are uploaded as 'Supporting Information' and are invariably referred to (in the manuscript, figure legends, and the Description field when uploading your files) using the following format verbatim: S1 Data, S2 Data, etc. Multiple panels of a single or even several figures can be included as multiple sheets in one excel file that is saved using exactly the following convention: S1_Data.xlsx (using an underscore).

b. Deposition in a publicly available repository. Please also provide the accession code or a reviewer link so that we may view your data before publication.

>>Regardless of the method selected, please ensure that you provide the individual numerical values that underlie the summary data displayed in the following figure panels as they are essential for readers to assess your analysis and to reproduce it:

FIGURES:

2BEG

4BD

5AC

6BEG

S2AB

S3

S5ABC

>>Please also ensure that figure legends in your manuscript include information on where the underlying data can be found, and ensure your supplemental data file/s has a legend.

>>Please ensure that your Data Statement in the submission system accurately describes where your data can be found.

5) SUPPLEMENTAL METHODS AND RESULTS: I noticed your paper has some of the methodological details and results in the supplement. Please move these to the main text of your manuscript. Note, we do not have a length requirement, and think it will make these details more accessible to our readership.

6) RCT DETAILS: I have a few requests related to the reporting of your clinical trial data, to bring your study in line with our clinical trials reporting policy (the full details of which can be found here:

https://journals.plos.org/plosbiology/s/human-subjects-research#loc-clinical-trials). Specifically -

a. We require that the CONSORT flow diagram be presented as a main figure (I see you have provided this as supplemental figure 1). Ideally the CONSORT diagram would be presented as figure 1 in the manuscript but if it is more natural to have it as a later figure, we are OK with that, as long as it appears the first time the RCT is mentioned.

b. Please provide the complete trial protocol related to your study as a supplementary file.

c. Please update your manuscript file to include the following information:

--An explanation of any deviation from the trial protocol [IF ANY, if none, please indicate that]

--Any information on statistical methods or participants not indicated in the CONSORT documentation [IF ANY]""

7) CODE: Per journal policy, if you have generated any custom code during the course of this investigation, please make it available without restrictions. Please ensure that the code is sufficiently well documented and reusable, and that your Data Statement in the Editorial Manager submission system accurately describes where your code can be found.

We expect to receive your revised manuscript within two weeks.

*Published Peer Review History*

*Press*

Sincerely,

Luke

Lucas Smith, Ph.D.

Senior Editor

lsmith@plos.org

PLOS Biology

---

## [Editor Report · Decision Letter 4]

31 Oct 2025

Dear Dr Knudsen,

Thank you for the submission of your revised Research Article "Sleep deprivation and sleep intensity exert distinct effects on cerebral vasomotion and brain pulsations driven by the respiratory and cardiac cycles" for publication in PLOS Biology and thank you for addressing our last editorial requests. On behalf of my colleagues and the Academic Editor, Pierre-Hervé Luppi, I am pleased to say that we can in principle accept your manuscript for publication, provided you address any remaining formatting and reporting issues. These will be detailed in an email you should receive within 2-3 business days from our colleagues in the journal operations team; no action is required from you until then. Please note that we will not be able to formally accept your manuscript and schedule it for publication until you have completed any requested changes.

PRESS

Sincerely, 

Lucas Smith, Ph.D.

Senior Editor

PLOS Biology

lsmith@plos.org